# Fuzzy recognition by the prokaryotic transcription factor HigA2 from *Vibrio cholerae*

San Hadži [1,2,3], Zala Živič[3], Matic Kovačič [4], Uroš Zavrtanik [3], Sarah Haesaerts[1,2], Daniel Charlier [5], Janez Plavec [4], Alexander N. Volkov[1,2,6], Jurij Lah [3] ✉ & Remy Loris [1,2] ✉

Disordered protein sequences can exhibit different binding modes, ranging from well-ordered folding-upon-binding to highly dynamic fuzzy binding. The primary function of the intrinsically disordered region of the antitoxin HigA2 from *Vibrio cholerae* is to neutralize HigB2 toxin through ultra-high-affinity folding-upon-binding interaction. Here, we show that the same intrinsically disordered region can also mediate fuzzy interactions with its operator DNA and, through interplay with the folded helix-turn-helix domain, regulates transcription from the *higBA2* operon. NMR, SAXS, ITC and in vivo experiments converge towards a consistent picture where a specific set of residues in the intrinsically disordered region mediate electrostatic and hydrophobic interactions while "hovering" over the DNA operator. Sensitivity of the intrinsically disordered region to scrambling the sequence, position-specific contacts and absence of redundant, multivalent interactions, point towards a more specific type of fuzzy binding. Our work demonstrates how a bacterial regulator achieves dual functionality by utilizing two distinct interaction modes within the same disordered sequence.

Since Emil Fisher in 1894 established the lock-and-key model for enzyme-substrate recognition, specificity in macromolecular recognition has been attributed to a combination of shape and chemical complementarity that requires folded conformations, at least in the bound state. Almost 15 years ago, in a landmark review, Peter Tompa and Monika Fuxreiter coined the term fuzzy complexes for instances where specific and often high-affinity macromolecular complexes are formed while at least one of the partners remains (partially) unfolded[1]. Since then, it has become clear that, like the intrinsically disordered proteins (IDPs) involved in this (at first glance irrational) specific recognition mechanism, fuzzy interactions come in many flavors. Fuzziness in macromolecular complexes ranges from discrete alternative binding modes and disordered segments that link ordered recognition elements to binding modes where one or both partners remain entirely disordered.

One mechanism of fuzzy recognition involves a large number of weak binding motifs, such as phosphate groups, resulting in a macroscopic binding constant that depends on the amount of such motifs present. Only a single motif interacts at any instance with the target protein, and the remainder of the IDP remains unfolded. An example here is the interaction between disordered yeast cyclin-dependent kinase inhibitor Sic1 and its receptor F-box protein Cdc4. The latter recognizes individual phosphate groups on the fully unfolded Sic1, and the overall affinity is generated via a re-binding

[1]Structural Biology Brussels, Department of Biotechnology, Vrije Universiteit Brussel, Pleinlaan 2, 1050 Brussels, Belgium. [2]Centre for Structural Biology, VIB, Pleinlaan 2, 1050 Brussels, Belgium. [3]Department of Physical Chemistry, Faculty of Chemistry and Chemical Technology, University of Ljubljana, 1000 Ljubljana, Slovenia. [4]Slovenian NMR Center, National Institute of Chemistry, Hajdrihova, 19, 1000 Ljubljana, Slovenia. [5]Research group of Microbiology, Department of Biotechnology, Vrije Universiteit Brussel, Pleinlaan 2, 1050 Brussels, Belgium. [6]Jean Jeener NMR Centre, Vrije Universiteit Brussel, Pleinlaan 2, 1050 Brussels, Belgium. ✉e-mail: jurij.lah@fkkt.uni-lj.si; remy.loris@vub.be

effect ensuing from the high local concentration of Sic1 phosphate groups presented to Cdc4[2].

Another class of fuzzy interactions encompasses two or more IDPs dancing around each other. This was first observed for the high-affinity interaction between histone H1 and the chaperone prothymosin-α[3]. Prothymosin-α interacts with the lysine-rich IDP region of histone H1 purely via electrostatic interactions without requiring defined binding sites or specific interactions between individual residues. Similar mechanisms lay also at the basis of the formation of membrane-less organelles via liquid-liquid phase separation[4]. Here, interactions among the macromolecule components often involve electrostatic interactions, but also cation-π, aromatic stacking or dipole-dipole interactions (for a review, see Brangwynne et al.)[5]. Patterning and valence are the main features that determine whether specific interactions leading to phase separation can be established[6,7].

More complex fuzzy recognition involves a central specific motif that folds upon binding, but its affinity is further enhanced via non-folding flanking regions that provide non-specific yet stabilizing interactions. Examples of the latter are found in the eukaryotic co-activator CREB-binding protein for the binding of its disordered KID domain to the globular KIX domain[8] and in the transcription activator Gcn4 binding to the mediator subunit Gal11[9,10]. In the latter, the interaction with the central element remains fuzzy as well due to multiple distinct binding modes[11]. This type of fuzziness has also been studied from a more theoretical and thermodynamic perspective but still remains poorly understood[12,13].

While IDPs have been studied mainly in higher eukaryotes, where they are particularly abundant, they are far from absent in prokaryotes and viruses[14-16], albeit less ubiquitous. Recent work has indicated that prokaryote IDPs are equally adept at carrying out complicated mechanistic tasks and integrating different functions, although mechanisms and functions seem to differ between eukaryote and prokaryote IDPs. For example, there is currently no data that suggests enrichment in post-translational modifications of prokaryote IDPs. IDPs from prokaryotes remain heavily understudied, as is also evident from the observation that prokaryote IDPs are significantly enriched in proteins that have no functions assigned in the Clusters of Orthologous Groups (COG) database[14].

Among the most extensively studied prokaryotic IDPs are several representatives of antitoxins from so-called toxin-antitoxin (TA) modules. Ubiquitous in bacterial genomes, these two-component systems are believed to stabilize mobile genetic elements (e.g., plasmids and integrons) as well as non-essential segments on chromosomes, contribute to abortive bacteriophage infection, and/or play a role in bacterial stress response (and possibly in the establishment or maintenance of the persister phenotype)[17]. Antitoxins typically consist of an ordered DNA binding domain coupled to a disordered region that has a toxin-neutralizing function and folds upon binding to the toxin (for a review, see Loris & Garcia-Pino[18]; De Bruyn et al.)[18,19]. In some instances, the function of IDP antitoxins extends from their main activity as a toxin inhibitor. An example here is the IDP-driven entropic exclusion observed when two PhD transcription factors bind to *phd/doc* operator in the absence of Doc[20] or the rejuvenation of CcdB-poisoned Gyrase[21]. The *higBA2* TA system from *Vibrio cholerae* consists of an mRNAse toxin HigB2 and its antitoxin inhibitor HigA2. Antitoxin HigA2 forms a tight complex with HigB2 through its N-terminal disordered region (IDR), which folds upon binding[22]. Here we show that in addition to the primary function of HigA2 IDR to act as a toxin inhibitor, the IDR also contributes to specific and high-affinity recognition of the *higBA2* operator to regulate its transcription. The HigA2 folded domain confers specificity towards a single inverted repeat operator sequence, while the IDR enhances its affinity. Using a combination of structural and biophysical techniques, we characterize the fuzzy IDR-operator interaction and discuss how two different binding modes can be encoded within the same IDR sequence of a prokaryotic regulator.

## Results

### A single operator site regulates the transcription of the higBA2 module

To gain insight into transcription regulation of the *Vibrio cholerae higBA2* module, we mapped the operator region using in-gel copper phenanthroline footprinting (Fig. 1a). In the 140 base pair (bp) DNA fragment, covering the region 85 bp upstream and 55 bp downstream of the *higB2* transcription start site, we identified a 24 bp long region that is protected when incubated with the antitoxin HigA2 (Fig. 1b). The protected region harbors a perfect 7 bp inverted repeat with a centrally located spacer TGTACGC(N)$_5$GCGTACA. The operator sequence lies upstream of the *higB2* gene, starting after the −35 promoter element and extending over the −10 element. Using electrophoretic migration shift assay (EMSA) we monitored the interaction of HigA2 with a 140 bp DNA fragment containing the operator site. At concentrations of HigA2 below 0.5 μM, we observe a shift of the operator fragment, indicating the formation of a specific protein–DNA complex (Supplementary Fig. 1). In contrast, for the fragment covering the region between the *higB2* and *higA2* genes, we observe only non-specific HigA2 binding at concentrations >10 μM, which is in a similar range as the binding to the unrelated DNA fragment (Supplementary Fig. 1).

### Only the globular domain of HigA2 is visible in its complex with the operator fragment

To understand the operator specificity of HigA2, its structure in complex with a 17 bp blunt end operator fragment was determined by X-ray crystallography (Supplementary Table 1). We have previously shown that HigA2 is a dimer in solution[22,23]. In the structure with the operator, HigA2 binds to the DNA also as a dimer and induces a 33° bend in the DNA molecule. (Fig. 1c). Only the globular C-terminal HigA2 domain, corresponding to the residues 37-104, could be observed; the N-terminal IDR remains disordered and lacks electron density. HigA2 belongs to the HTH-XRE family of DNA binding proteins and binds each half-site of the operator through its HTH motif formed by helices α2 and α3, while helices α4 and α5 form the HigA2 dimerization interface (Fig. 1c). Specific base recognition is mediated by four amino acid residues of the recognition helix α3: Arg68, Glu71, Asn72, and Arg77, with the latter adopting a double conformation. These residues interact with both strands of the half-palindrome. In its first conformation, the side chain of Arg77 forms a hydrogen bond with O6 atoms of the guanine 5 and guanine 12 on the opposing DNA strands (Supplementary Fig. 2). In its alternative conformation, the side chain of Arg77 interacts with the phosphate backbone. The side chain of Glu71 forms a hydrogen bond to N6 of the adenine 3 on strand 1, while Arg68 and Asn72 contact N7 atoms of the guanines 14 and 12 on strand 2. Surrounding the base-specific recognition sites, there are additional interactions with the phosphate backbone involving side chains of eight other amino acid residues, which are shown schematically in Fig. 1d. In addition to structural changes of the DNA molecule, HigA2 slightly contracts upon DNA binding (Supplementary Fig. 2). Compared to the free HigA2, the distance between the two recognition helices α3 decreases in order to match the separation of the major groove segments.

### The HigA2 intrinsically disordered domain enhances operator binding

To elucidate the role of the disordered domain on the HigA2-operator interaction, we prepared a truncated variant HigA2$_{\Delta IDR}$, corresponding to the globular C-terminal domain for which electron density is observed in our crystal structure (protein sequences are listed in Supplementary Table 2). HigA2$_{\Delta IDR}$ is folded and has the same thermodynamic stability as the full-length protein (Supplementary Fig. 3). Surprisingly, the operator binding affinity measured by isothermal titration calorimetry (ITC) is significantly reduced upon removal of the

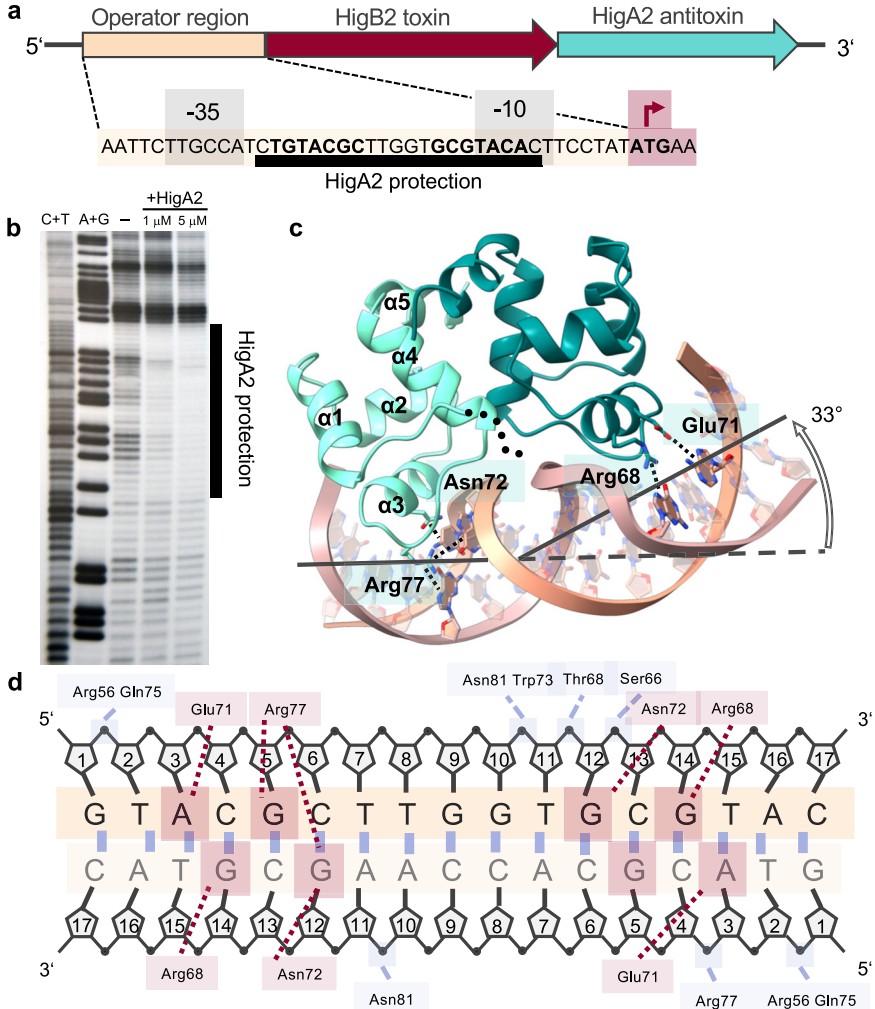

**Fig. 1 | Crystal structure of the HigA2 in complex with its operator.**
**a** Organization of the *higBA2* operon. Transcription is regulated through the operator region containing an inverted repeat (bold), identified by a DNA footprinting. Gray boxes show the −35 and −10 sites. **b** In-gel copper-phenanthroline footprint of the 140 bp DNA fragment upstream of *higBA2*. Free (−) and bound DNA (+HigA2) were separated by EMSA, treated in gel with the copper-phenanthroline ion, and the reaction products were separated by gel electrophoresis in denaturing conditions. A + G and C + T are the corresponding chemical sequencing ladders.

**c** Crystal structure of the HigA2 dimer C-terminal domain (residues 37–104) in complex with the 17 bp operator fragment. N-terminal residues 2–36 from the protein are presumably disordered and indicated as black dotted lines. Four residues mediating base-specific contacts are shown as sticks, and only one pair per monomer is shown for clarity. **d** Schematic overview of the HigA2 C-terminal domain interactions with the operator. Red lines represent hydrogen bonds to nucleic bases, blue lines are interactions with the phosphodiester backbone.

disordered domain (Fig. 2a). The full-length HigA2 binds the 45 bp DNA fragment containing operator site (Opr45, DNA sequences are listed in Supplementary Table 3) with high affinity ($K_D = 25.1 ± 5.2$ nM). In contrast, a more than 15-fold decrease in affinity is observed for HigA2$_{ΔIDR}$ ($K_D = 400 ± 50$ nM), indicating that the N-terminal IDR domain contributes to operator binding (Table 1). A titration of HigA2 into a shorter DNA fragment (Opr25) gives identical affinity as with the longer fragment (Opr45) suggesting that IDR interactions do not extend much further from the central binding motif (Table 1, Supplementary Fig. 4). A peptide corresponding to the IDR (HigA2$_{IDR}$) does not bind the operator by itself (Supplementary Fig. 4), suggesting that the globular domain not only recognizes the operator but also optimally positions IDR relative to the operator to enable interactions.

To delineate the nature of interactions between the intrinsically disordered domain and the DNA operator, we analyzed the thermodynamics of the binding of HigA2 and HigA2$_{ΔIDR}$ to Opr45. The removal of the disordered domain decreases the entropic penalty, indicating that the IDR acts as an entropic barrier due to conformational

restriction of the ensemble (Fig. 2b). At the same time, the favorable enthalpic contribution is reduced by half, indicating a loss of attractive interactions mediated by the disordered domain (Fig. 2b). The binding enthalpy decreases linearly with temperature, indicating a negative heat capacity change (Fig. 2c), which scales with the amount of hydrophobic surface buried upon binding[24]. Given that HigA2$_{ΔIDR}$ has a lower heat capacity (lower slope), this suggests that removal of the IDR also removes hydrophobic contacts mediated by the IDR.

To evaluate the effect of the IDR domain on transcription in vivo, we designed a reporter vector based on the red fluorescent protein (mRFP1) regulated under the *higBA2* operator. Cells harboring this vector have strong fluorescence intensity per optical density, indicating that mRFP1 is synthesized from the *higBA2* promotor (Fig. 2d). However when HigA2 under a constitutive promotor is added to this vector, we observed a decrease in the normalized fluorescence intensity, indicating repression of mRFP1 transcription. In contrast, expression of the truncated antitoxin HigA2$_{ΔIDR}$ does not repress mRFP1 transcription, in line with our in vitro results (Fig. 2d). As a control, we tested the

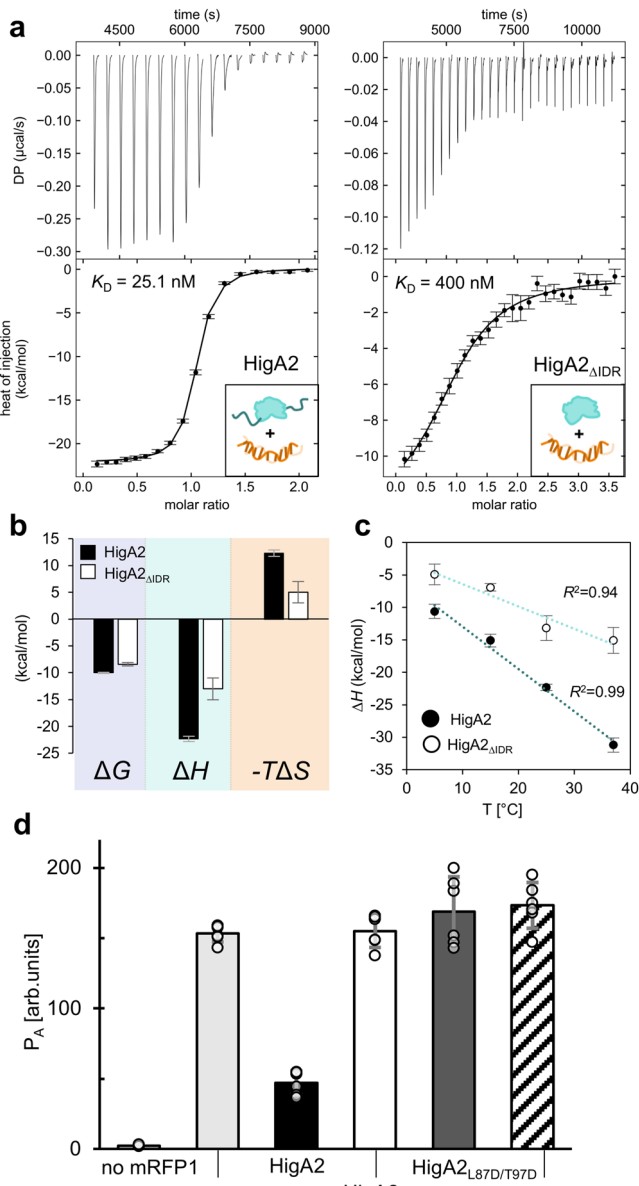

**Fig. 2 | The N-terminal disordered domain of HigA2 interacts with the operator DNA. a** Interaction of HigA2 and the HigA2$_{\Delta IDR}$ variant lacking N-terminal disordered domain with the operator DNA. The top panels show a corrected heat rate, and the bottom panels have normalized heat. Fits of the single-site binding model function are shown as solid lines. **b** Thermodynamic profile of HigA2 and HigA2$_{\Delta IDR}$ binding to DNA at 25 °C. Bars show best-fit model parameters (Table 1) for the binding experiments shown above ($n = 1$) with error bars showing one s.d. **c** Dependence of the binding enthalpy on temperature for HigA2, and HigA2$_{\Delta IDR}$ shows a difference in heat capacity change upon operator binding. Error bars show one s.d. **d** In vivo repression assay. Bars show the *higBA2* promotor activity based on the fluorescence intensity of *E. coli* cells with reporter vectors expressing mRFP1 under the *higBA2* operator. When HigA2 under weak constitutive promotor is added to the vector, expression of mRFP1 is reduced, while variants HigA2$_{\Delta IDR}$, HigA2$_{IDR}$, and HigA2$_{L87D/T97D}$ show no repression. Bars show the mean value from $n = 6$ independent experiments with errors corresponding to one s.d.

repression of IDR alone (HigA2$_{IDR}$) and HigA2 mutant with impaired dimerization of the globular domain (HigA2$_{L87D/T97D}$). Neither of these proteins represses transcription, therefore, functional repression is achieved only via the combined interplay of globular and IDR domains (Fig. 2d).

## Shuffling of the IDR sequence abolishes operator binding and acts only as an entropic barrier

To understand the level of specificity encoded in the disordered region, we prepared a variant HigA2$_{Shuff}$ where the sequence of the disordered tail is shuffled (Supplementary Table 2). In contrast to the tolerance to shuffling that is often seen in functional IDPs[25,26], we observe that sequence shuffling significantly decreases the affinity for Opr25 ($K_D = 37 \pm 20$ µM) (Table 1, Supplementary Fig. 4). Strikingly, the affinity of HigA2$_{Shuff}$ for Opr25 is even weaker than for the HigA2$_{\Delta IDR}$ truncate, where the disordered domain was removed. This suggests that the disordered tail with the shuffled sequence loses the ability to establish favorable interactions with the DNA but acts only as an entropic barrier, thus reducing binding affinity.

To investigate whether the disordered tails from other antitoxin repressors could also interact with the operator, we prepared a chimeric variant harboring the HigA2 globular domain and the disordered domain of the Phd antitoxin (Supplementary Table 2). Similarly to the shuffled variant, chimeric HigA2$_{Phd}$ binds the operator with low affinity ($K_D = 10 \pm 5$ µM), indicating that the operator interactions are specific for the HigA2 IDR sequence and that the presence of generic disordered variant decreases operator affinity likely by acting as an entropic barrier (Table 1, Supplementary Fig. 4).

## The IDR residues contacting DNA are located at specific positions along the disordered domain

To map the DNA interactions onto the IDR sequence, we designed three HigA2 variants, each having 12 consecutive residues replaced by a tract of small polar amino acids (serine, threonine, glycine, and alanine), which we refer to as HigA2$_{Mut(2-13)}$, HigA2$_{Mut(14-25)}$ and HigA2$_{Mut(26-37)}$ (Supplementary Table 2). These HigA2 variants unfold cooperatively and display similar thermal stability to wild-type HigA2 (Supplementary Fig. 3). All three variants bind the operator Opr25 with lower affinity than the wild-type protein, indicating that the introduced mutations disrupt protein-DNA interactions (Table 1, Supplementary Fig. 4). The drop in affinity increases from the N-terminus onwards: the HigA2$_{Mut(2-13)}$ variant shows over 20-fold decrease in affinity ($K_D = 0.39 \pm 0.13$ µM), HigA2$_{Mut(14-25)}$ a 300-fold decrease ($K_D = 5.1 \pm 2.4$ µM), while HigA2$_{Mut(26-37)}$ variant shows no binding to Opr25 (Table 1).

To further delineate IDR-DNA contacts, we performed alanine scan mutagenesis and evaluated the repressor activity of HigA2 variants in vivo using the fluorescence of mRFP1 as a readout. Most alanine mutants that lose their repressor activity are clustered between the residues 24 and 32, in proximity to the folded domain, while few mutants are located near the N-terminus (Fig. 3). They include Lys24, Leu25, Thr26, Lys28, Thr29, and Val32, and the N-terminal Asn3 and Leu6. In addition, partial loss of repression is observed for Leu27 and Ser2, which are also part of these two residue clusters. Thus, both electrostatic and hydrophobic residues seem to mediate IDR-DNA contacts, although next to direct interactions, these mutations may also affect the structural ensemble and modulate the conformation of nearby residues. Of these residues, Lys24, Thr26, Leu27, and partially also Lys28 are conserved in HigA2 homologs, while Leu25 is variable (Supplementary Fig. 5). Leu27, Lys28, and Val32 are involved in contacts with the HigB2 toxin, but they are not essential for toxin neutralization[27], suggesting that the evolutionary conservation may be related to fuzzy DNA binding.

Several alanine mutants show increased repression, most of which concern negatively charged amino acids: Glu9, Glu16, Glu22, and Asp34 (Fig. 3). This indicates that charge repulsion influences DNA binding, however, also Gln19, Gly23, and Asn33 are suboptimal for obtaining the strongest possible repression. Except for Glu9, none of these are involved in the interaction with HigB2[22], eliminating constraints originating from the HigB2–HigA2 interaction as an

**Table 1 | Standard thermodynamic parameters of binding obtained from fitting the single-site binding model function to the ITC titration curves**

| $T = 25\,°C$ | $K_D$ (nM) | $\Delta G°$/kcal mol$^{-1}$ | $\Delta H°$/kcal mol$^{-1}$ |
|---|---|---|---|
| HigA2 into Opr45 | 25.1 ± 5.2 | −10.3 ± 0.1 | −22.3 ± 0.5 |
| HigA2$_{\Delta IDR}$ into Opr45 | 400 ± 50 | −8.7 ± 0.3 | −13.2 ± 1.9 |
| HigA2$_{IDR}$ into Opr45 | No binding | / | / |
| HigA2 into Opr25 | 13.8 ± 5.8 | −10.6 ± 0.3 | −22.7 ± 1.8 |
| $T = 5\,°C$ | | | |
| HigA2 into Opr25 | 15 ± 3 | −11.2 ± 0.2 | −10.1 ± 1.1 |
| HigA2$_{\Delta IDR}$ into Opr25 | 480 ± 160 | −8.0 ± 0.4 | −5.2 ± 0.3 |
| HigA2$_{Mut(2-13)}$ into Opr25 | 390 ± 130 | −8.2 ± 0.2 | −11.8 ± 1.3 |
| HigA2$_{Mut(14-25)}$ into Opr25 | 5,100 ± 2400 | −6.7 ± 0.6 | −7.2 ± 2.0 |
| HigA2$_{Mut(26-37)}$ into Opr25 | No binding | / | / |
| HigA2$_{Shuff}$ into Opr25 | 37,000 ± 20,000 | −5.6 ± 0.6 | −29.4 ± 2.3 |
| HigA2$_{Phd}$ into Opr25 | 10,000 ± 5000 | −6.3 ± 0.3 | −21.8 ± 2.1 |

Errors correspond to one s.d. and were calculated from the Monte Carlo analysis of the single-site binding model fits to the ITC data.

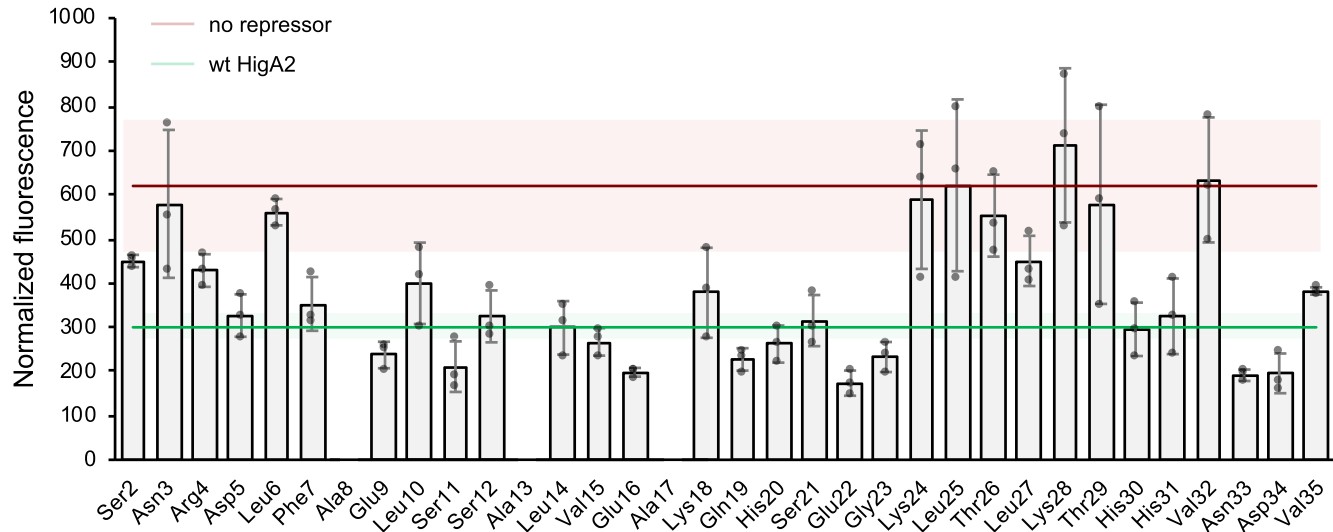

**Fig. 3 | Alanine scan mutagenesis of the HigA2 disordered domain.** Bars show the OD-normalized fluorescence intensity for strains expressing mRFP1 under *higBA2* promotor and different HigA2 alanine mutants. Bar values correspond to the mean values from $n = 3$ independent experiments with errors corresponding to one s.d. The red line shows the intensity of the unrepressed *higBA2* promotor (vector without HigA2), while the green line is the repressed *higBA2* promotor (vector with the wild-type HigA2). The corresponding colored regions show one s.d. from $n = 6$ independent experiments.

evolutionary compromise to retain possible sub-optimal repression. For a selection of seven alanine mutants we tested whether mutations compromise toxin neutralization, which would indicate the presence of functional trade-offs in IDR. Bacterial spotting assay using active HigB2 toxin and different HigA2 alanine mutants shows that selected single-residue alanine mutations do not compromise toxin neutralization (Supplementary Fig. 5). This suggests that the evolutionary conservation in the 24–32 residue cluster may be functionally related to the operator repression.

**The IDR remains disordered upon operator binding**

To probe the global structure of the HigA2-operator ensemble in solution, we used circular dichroism (CD) spectroscopy and small-angle X-ray scattering (SAXS). The CD spectrum of HigA2 is reminiscent of an all-alpha protein, in agreement with the structure of its C-terminal helix-turn-helix (HTH) containing DNA binding domain (Fig. 4a). Upon binding of HigB2, the helical content increases in agreement with folding-upon-binding of the N-terminal IDR domain as shown previously[22] (Fig. 4a). In contrast, upon operator binding only

marginal changes in the CD signal can be observed, confirming that the N-terminal IDR of HigA2 does not fold upon binding to the operator and remains disordered (Fig. 4a). To further assess the level of disorder we compared the dimensionless Kratky plots of free HigA2 and operator-bound HigA2. While free HigA2 shows typical features of a spherical particle with a disordered tail, the plot of the HigA2-operator complex is closer to what is observed for globular particles, suggesting reduced flexibility of the disordered tail (Fig. 4b).

Further structural insight into the HigA2 disordered domain and its interaction with the operator comes from solution NMR spectroscopy. Nearly complete assignments of protein backbone amide resonances (99 out of 103 residues) were obtained from standard three-dimensional NMR experiments on the $^{13}$C, $^{15}$N isotopically labeled HigA2 (Fig. 4c, Supplementary Fig. 6). For the majority of the Cα, Cβ, CO and Hα atoms in the N-terminal IDR, chemical shifts do not differ significantly from those of the random coil, except for residues 7–21, which appear to populate a transient helical structure (Supplementary Fig. 6), in agreement with the previously observed low helix contents of the HigA2 IDR peptides[27]. Titration of Opr25 into

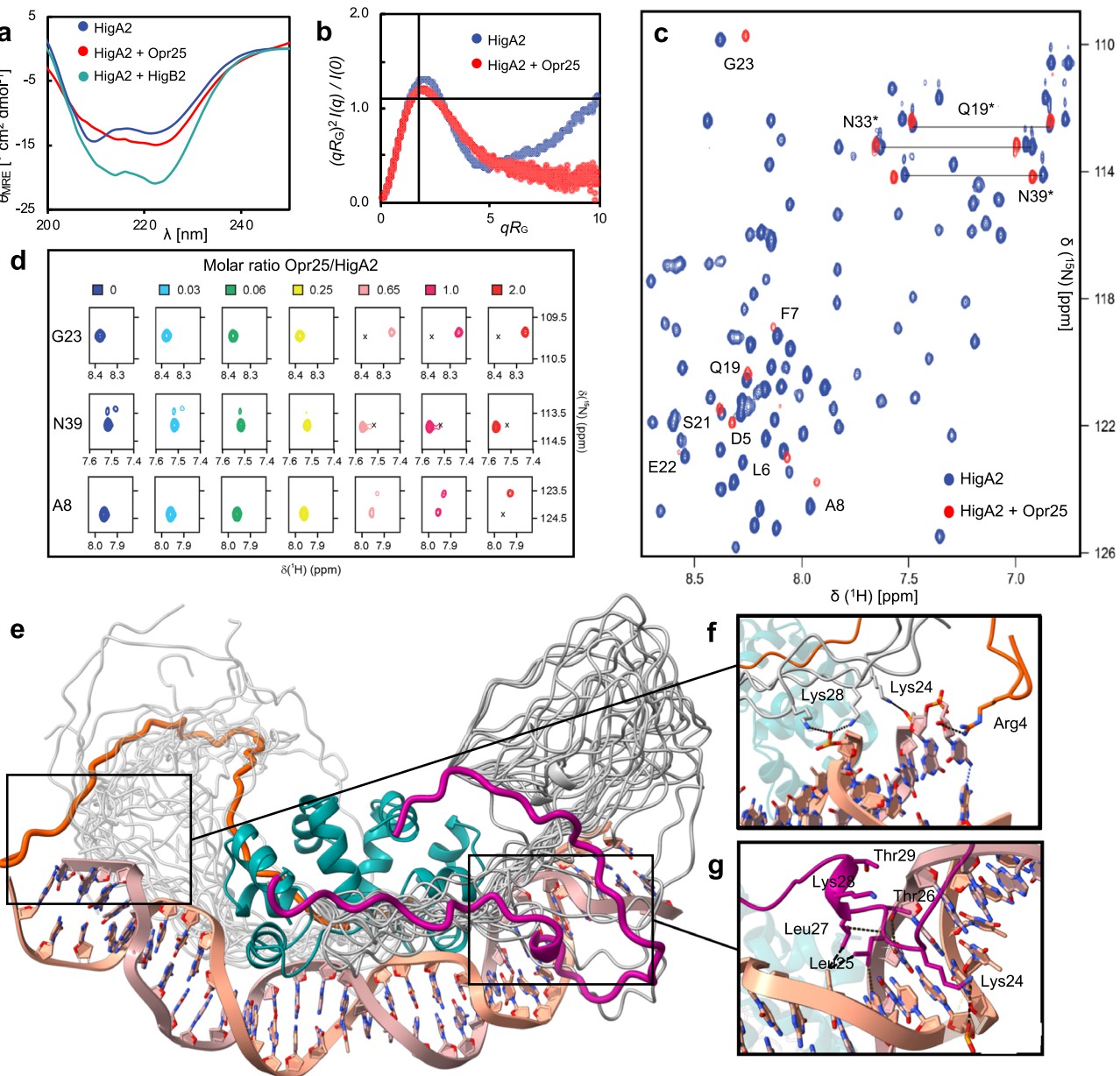

**Fig. 4 | Structural ensemble of the HigA2-operator fuzzy complex. a** Circular dichroism spectra of HigA2, and the difference spectra of HigA2 in complex with HigB2 or the operator. No significant gain of HigA2 secondary structure is observed upon operator binding, in contrast with binding to toxin HigB2. **b** Dimensionless Kratky plots indicate a reduction of the conformational ensemble of HigA2 in complex with the operator. The intercept of the gray lines (√3, 1.104) marks the position of the maximum for an ideal globular protein. **c** Overlay of the [$^1$H,$^{15}$N] HSQC spectra of HigA2 alone (blue) or in the presence of 2 molar equivalents of Opr25 (red). Resonances observed in the HigA2-operator complex are labeled. Sidechain amides of Q19, N33, and N39 are connected by horizontal lines and

indicated by asterisks. **d** The NMR titration of HigA2 with Opr25. [$^1$H,$^{15}$N] HSQC spectral regions for several representative resonances are plotted at different [DNA]/[HigA2] ratios. The positions of the free HigA2 peaks are indicated by the cross. **e** The 10-lowest energy structures of the HigA2–operator complex were obtained from the Xplor-NIH protocol. The globular domain is shown in cyan, disordered N-terminal domain is in gray. Representative models of two conformers and shown in orange and violet. **f** Close view of the selected electrostatic and **g** hydrophobic interactions with the DNA. In one of the ensemble models shown here, the IDR segment 24–29 docks into the minor DNA groove.

$^{15}$N-labeled HigA2 induces a progressive, general decrease of peak intensities (Fig. 4c, Supplementary Fig. 7) caused by an increase in the rotational correlation time due to the formation of a rather large (39 kDa) HigA2-Opr25 complex. Most of the cross-peaks in the [$^1$H, $^{15}$N] HSQC spectrum belonging to the residues in the HigA2 globular domain disappear at 0.65 molar equivalents of DNA (Supplementary Fig. 7). However, those from the disordered domain still remain visible at up to 2 equivalents of DNA (Fig. 4c), indicating that its ensemble remains highly dynamic. Some of the HSQC cross-peaks of the IDR segment progressively disappear and start to reappear elsewhere in

the spectrum during the titration (Asp5, Leu6, Phe7, Ala8, Gln19, Ser21, Glu22, Gly23, Fig. 4d). This behavior is typical for the NMR slow exchange regime and indicates a change in the chemical environment upon DNA binding. We could explicitly assign several of these cross-peaks in the bound form using triple-resonance experiments on the $^{13}$C, $^{15}$N HigA2–Opr25 complex (Fig. 4c). Interestingly, except for Leu6, these amino acids do not include the essential residues that were revealed by the alanine scan. It is likely that the residues with detected NMR binding shifts do not interact directly with the DNA. Rather they experience secondary binding effects propagating from the

neighboring groups, which pull them towards the DNA, thereby altering their chemical environment. Such indirect binding effects could explain why, unlike for residues involved in direct interactions with the DNA, the affected resonances remain visible in the DNA-bound form and account for the modest chemical shift changes seen in the HSQC spectrum (Fig. 4c). Overall, the observed NMR chemical shift perturbations confirm that the HigA2 N-terminal domain engages in direct interactions with the operator DNA.

## The IDR "hovers" over the operator, making transient contacts with DNA

To understand how IDR interacts with the operator in more detail, we built a model of the complex using the HigA2-operator crystal structure, while the flexible N-termini were modeled using molecular dynamics. The ensemble calculation was performed with an Xplor-NIH protocol that includes SAXS data as structural restraints and uses molecular ensembles of multiple conformers ($N = 1$–5) to find the best agreement with the experimental data. A molecular ensemble of two conformers gives the best agreement with the experimental scattering profile ($\chi^2 = 1.19$) (Fig. 4e, Supplementary Fig. 8). The ten lowest-energy solutions generally fit within the molecular envelopes calculated from ab initio models with DAMMIF (Supplementary Fig. 8).

To examine the global structural characteristics of the HigA2-operator ensemble, we calculate the minimal distances between HigA2 C$\alpha$ atoms and phosphorous atoms of the DNA backbone for each conformer (Supplementary Fig. 8). For the first, larger group of conformers, IDR residues 25-37 run parallel to the DNA helix at a distance of ~5–10 Å from the DNA backbone, while the N-terminal part of the chain extends out of the globular core (Fig. 4e, a representative model is shown in violet). This set of conformers is consistent with our ITC, alanine scan, and NMR data, which reveal the importance of the segment between residues 24–32 for operator binding. For the second group of conformers, we observe a different global conformation of the disordered domain. Initially, the chains extend away from the DNA and then loop back to make contacts via their N-terminal segments (Fig. 4e, a representative model shown in orange). This again agrees with our NMR and alanine scan data where we observe chemical shift perturbations upon addition of DNA and loss of repressor activity upon mutation to Ala for some of the residues near the N-terminus (e.g., Asn3, Leu6, Ala8) (Figs. 3 and 4). Examination of individual models reveals transient interactions between IDR and DNA backbone, many of which are mediated by the residues picked up in Ala scan (Fig. 4f). In one of the models the segment between residues 24 to 29 fits into the DNA major groove mediating both charged (Lys24 to phosphate backbone) and hydrophobic contacts (Leu25 and Leu27 to thymine methyl group) (Fig. 4g). This agrees with the mutagenesis data where mutations in this segment strongly decreases HigA2 repression (Fig. 3). Thus, as derived from the SAXS analysis, ensembles of two conformers concomitantly sampled in solution summarily account for the experimental results obtained here by other, complementary techniques, including NMR, ITC, and alanine scanning assays of the repressor activity.

## Discussion

Since the coining of the term fuzzy complexes 15 years ago[1], it has become clear that fuzzy recognition between macromolecules is a widespread phenomenon with implications for disease[28–31]. However, our understanding of this phenomenon remains limited as its structural and thermodynamic basis remains unclear. Here we dissect the fuzzy recognition of a DNA target by the intrinsically disordered tail of a bacterial transcription regulator, *V. cholerae* HigA2. The HigA2 C-terminal DNA binding domain provides specificity for a particular operator sequence, while the disordered domain considerably strengthens this interaction. Using a combination of structural and biochemical techniques, we show that the IDR of HigA2 uses specific

amino acid side chains to make transient interactions with its operator while hovering over the DNA duplex. This corresponds to a second binding functionality of this IDR next to its folding-upon-binding to toxin HigB2[22].

The degree of IDR-operator specificity that is present in the HigA2 system is unexpected. In general, disordered proteins are known to be resistant to the shuffling of their sequences. This is particularly prominent in transcription factors and other DNA-interacting proteins from eukaryotes. For example, the function of the IDRs of linker histones seems to depend essentially on their specific amino acid composition rather than on a specific sequence order of amino acids[26]. Similarly, long disordered regions in yeast transcription factor Msn2 and Yap1 are necessary to recognize target promotors but are resistant to sequence shuffling and partial truncations[32]. Other well-known examples of nonspecific, electrostatically-driven enhancement of DNA binding are positively charged tails of HGM-boxes and AT-hooks[33,34]. Most prominent in this context are transcription activation domains (TADs). In the classic mechanism of transcription activation, TADs bind to a variety of partners using an array of loosely defined multiple transient interactions[25,35]. Sequence independence, lack of consensus motifs, and the apparent potential of many unrelated TADs to interact with a large number of transcription mediators result in serious difficulties in explaining the mechanisms of action of TADs, making this a major unresolved question in modern biochemistry[36].

The default behavior of an IDR segment attached to a globular domain is to prevent its association with potential interaction partners or at least strongly weaken this interaction (Fig. 5a). As illustrated here, binding requires the IDR ensemble to become more compact. This likely involves a significant entropic cost associated with a reduced space that is left available for the IDR without the compensating enthalpic contributions associated with folding-upon-binding. Indeed, this phenomenon is observed for HigA2 when its native IDR is replaced by a non-native one. Both a version with a shuffled IDR sequence or its mimic with a sequence from an unrelated antitoxin (Phd) leads to a large affinity decrease for the operator compared to that of the individual folded domain of HigA2. This finding is in line with regulatory mechanisms employed by other toxin-antitoxin systems where an IDR stretch is used to weaken operator binding[20,37]. It also mirrors the mechanisms otherwise seen in eukaryote IDPs such as entropic bristles, which regulate transport through the nuclear pore complex[38], drive membrane curvature[39], modulate the equilibrium between substates of human UDP-α-D-glucose-6-dehydrogenase[40] or change aggregation behavior of prion sequences[41].

The HigA2 wild-type IDR sequence is able to mediate sufficiently strong interactions with the DNA to overcome the IDR conformational penalty (Fig. 5b), however, if these transient interactions are disrupted, the affinity decreases substantially, and entropic exclusion takes over. The overall enthalpic contribution stems from several relatively larger interaction clusters rather than from many weaker interactions that are uniformly distributed over the sequence. This contrasts the commonly observed situation in fuzzy complexes that rely on multiple, weak, glue-like interactions. We have previously analyzed the thermodynamics of the HigA2−HigB2 folding-upon-binding interaction and observed that the binding interface is highly optimized in terms of enthalpy, reaching similar levels as in the tightest protein complexes like barnase-barstar[27]. Given that the HigA2−HigB2 binding affinity is in the picomolar range, such strong optimization of interactions is necessary to compensate for the IDR folding free energy penalty, which for folding-upon-binding interactions can be as high as +3.5 kcal/mol[42]. Even though the penalty arising from the restriction of the IDR conformational space, in principle, lowers binding affinity, it is conceivable that it also stimulates the formation of stronger interactions, leading to increased specificity. In this scenario, the entropic penalty takes the role of a selection filter which attenuates weaker interactions while sufficiently strong IDR interactions are retained.

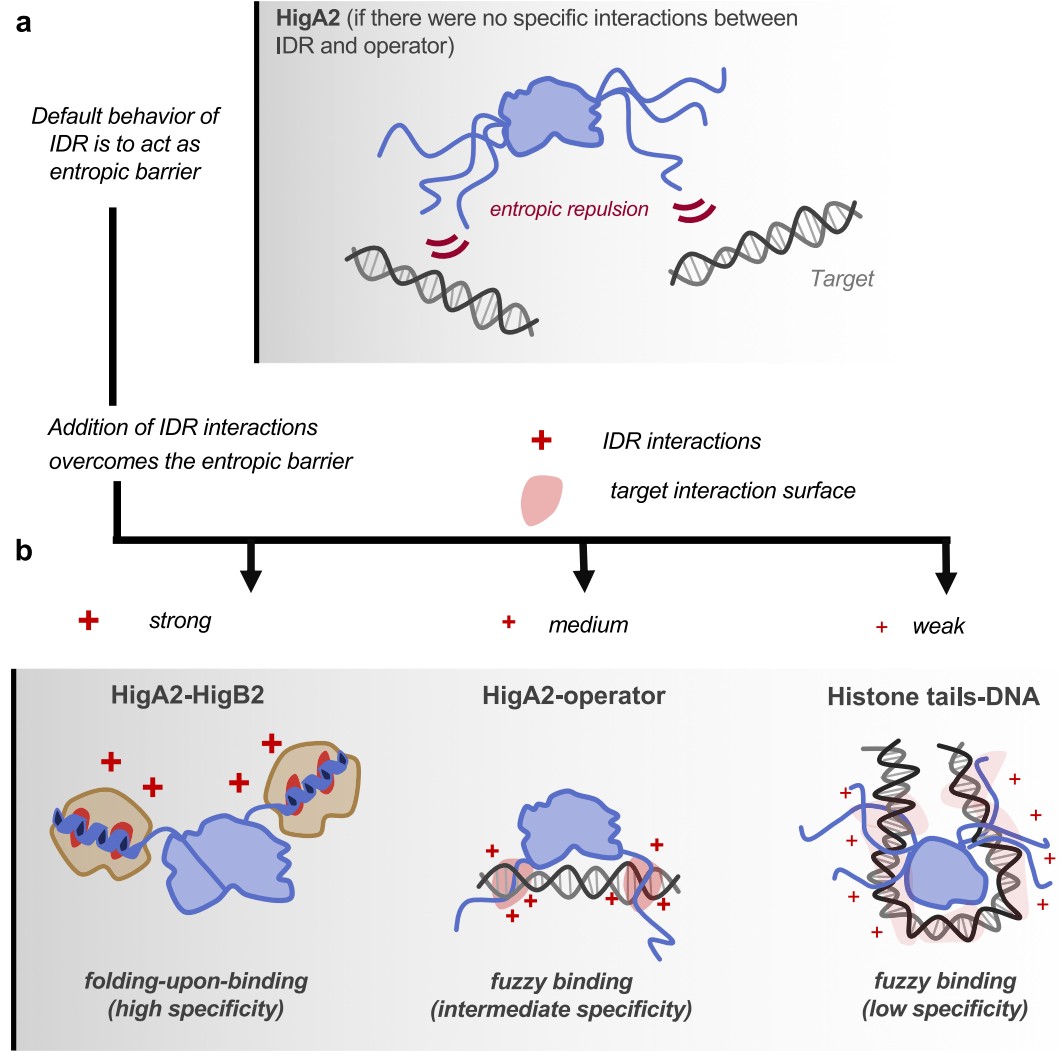

**Fig. 5 | Schemes of target recognition emphasizing different degrees of specificity and dynamics. a** A default behavior of an IDR, which is unable to mediate any target interactions, is to obstruct association with the target due to the entropic penalty coming from the restriction of the IDR conformational space. **b** Depending on the strength and the distribution of interactions on the IDR, a continuum of specificity and dynamics can be achieved. In the folding-upon-binding mechanism (left), as observed in the HigA2–HigB2 complex, a relatively small number of strong interactions are highly localized both at the level of IDR and the target, leading to high specificity. The opposite situation occurs in the fuzzy binding via multiple, weak, highly redundant interactions between histone tails and DNA (right). The intermediate level of specificity is observed in the HigA2-operator (middle), where interactions are localized at the IDR level but more broadly distributed at the DNA level.

The specificity observed in the IDR-DNA interaction of HigA2 positions somewhere between classic folding-upon-binding and a fully glue-like behavior that depends fully on non-specific electrostatic or hydrophobic attraction (Fig. 5b). We previously proposed a model for the sequence-ensemble relationship in fuzzy complexes that employs a simple distribution of target hotspots to which specific individual side chains of a partner IDR can dynamically dock[13]. This model extends classic folding-upon-binding towards a more dynamic version but essentially maintains the specificity within both IDR and target: the number of key interacting side chains of the IDR matches the number of hotspot pockets to which they dock (Fig. 5b, left). In the context of protein–DNA interactions, leucine zippers and basic helix-loop-helix domains are typical examples of folding-upon-binding, where strong sequence specificity is observed both at the level of IDR and the target DNA[43]. The HigA2-operator interaction represents a further deviation from the folding-upon-binding model in the sense that at the level of the IDR, specific amino acid side chains remain involved, and even single point mutations can abolish or significantly enhance binding in the absence of a folded conformation (Fig. 5b, middle). On the other

hand, the situation is different at the target: the DNA forms a continuous surface without specific hotspot pockets, or at least a surface with a much larger number of pockets which the IDR can choose from. Finally, the furthest away in our continuum from highly defined towards most fuzzy interactions will be those where the interacting partners fully glide over each other without any uniquely identifiable residues that determine specificity but still provide a functional interaction (Fig. 5b, right). Histone tails interacting with nucleosomal DNA may be such examples[26], where the enthalpic contribution is obviously sufficient to overcome the entropic cost to stabilize the interaction, even if this interaction is highly promiscuous.

HigA2 represents an illustrative example of a prokaryotic system utilizing key aspects of intrinsic disorder, which mirrors the functionality of disordered proteins from eukaryotic systems. On the one hand, the HigA2 IDR can engage in ultra-tight, picomolar association with HigB2 to neutralize its toxicity, whereby it folds into a well-defined helical structure with a highly specific, energy-optimized binding interface[27]. On the other hand, we show that this IDR also contributes to the selective recognition of the operator and enhances the

HigA2-operator association via fuzzy interactions with intermediate specificity. It is fascinating how these two structurally and thermodynamically fundamentally different molecular recognition modes are encoded into a single, short IDR segment.

## Methods

### Cloning and mutagenesis

All oligonucleotides (see Supplementary Table 3) and chemicals were purchased from Sigma-Aldrich. Gene sequences corresponding to the HigA2, HigA2$_{\Delta IDR}$, HigA2$_{Shuff}$, HigA2$_{Phd}$, HigA2$_{Mut(2-13)}$, HigA2$_{Mut(14-25)}$, and HigA2$_{Mut(26-37)}$ proteins (see Supplementary Table 2) were chemically synthesized and cloned into pET21b expression vector using NdeI and XhoI restriction sites by a commercial supplier (Genscript). In these constructs, an un-cleavable histidine tag is placed at the protein C-terminus. Gene region covering the *higBA2* operator region (starting 156 bp prior to the *higBA2* transcription/translation start site and 40 bp after it) was chemically synthesized and cloned into pUC57 using Xba I and BamH I restriction sites by a commercial supplier (Genscript). For the in vivo reporter system, we used the pSB1A3 vector harboring the red fluorescent protein mRFP1 under the P$_{higBA2}$ promotor (vector pSB1A3-P$_{higBA2}$-RFP). Gene fragments containing HigA2, HigA2$_{\Delta IDR}$, HigA2$_{L87D/T97D}$, and HigA2$_{IDR}$ with the upstream constitutive promotor P$_{117}$ from the Anderson promotor library (BioBrick identifier BBa_J23117)[44] were then ordered from a commercial supplier (Genescript). These fragments were inserted into the pSB1A3- P$_{higBA2}$-RFP using NdeI and PstI sites. All constructs were verified by gene sequencing using oligonucleotides *pSB1A3-F* and *pSB1A3-R* (Supplementary Table 3).

### Oligonucleotide and protein purification

DNA duplexes were prepared by mixing equimolar amounts of single strands in triple distilled water, heating the mixture to 95 °C, and slowly cooling it down to room temperature. The formation of all double-stranded DNA was checked by gel electrophoresis. Synthetic peptide HigA2$_{IDR}$ was bought from ChinaPeptides and was at least 95% pure. Proteins HigA2, HigA2$_{\Delta IDR}$, HigA2$_{Shuff}$, HigA2$_{Phd}$, HigA2$_{Mut(2-13)}$, HigA2$_{Mut(14-25)}$ and HigA2$_{Mut(26-37)}$ were expressed in *E. coli* BL21[DE3] strain harboring pET21b plasmids. Cells were grown at 37 °C until optical density reached 0.6 and induced by the addition of 1 mM IPTG. After 4 h, cells were harvested and lysed in 50 mM Tris pH 8, 150 mM NaCl (Buffer A), and protease inhibitors. Proteins were then purified using Ni−Sepharose columns. After loading the columns with a cell lysate, residual protein fractions were eluted with Buffer A plus 500 mM imidazole using a step gradient. Fractions containing proteins were loaded into a Bio-Rad SEC 70 column equilibrated with 20 mM Tris pH 8, and 150 mM NaCl. Isotopically labeled proteins were purified using the same procedure, except that cells were grown in isotope-containing media optimized for high-performance protein expression (Silantes GmbH, Germany).

### DNA footprinting and EMSA

The 140 bp long fragment used for footprinting was amplified using the PCR with pUC57 as a template and labeled primers *higBA2* oprF and *higBA2* oprR (Supplementary Table 3). Primers were labeled with [γ$^{32}$P]-ATP (Perkin Elmer) and T4 polynucleotide kinase (Fermentas). Labeled DNA fragment was incubated with or without HigA2 in 20 mM Tris pH 8 and 200 mM sodium chloride for 1 h, separated using EMSA, and treated in gel with the copper-phenanthroline ion. The reaction products are separated by gel electrophoresis in denaturing conditions along with the corresponding sequencing ladders[45].

For EMSAs, 140 bp long fragments were amplified using Cy5 labeled primers *higBA2* inter, *higBA2* opr, and rand using the pet15b expression plasmid (containing *higBA2* gene region) or pUC57 (containing *higBA2* operator region) as templates. Protein−DNA complexes were prepared by mixing the 5′-labeled DNA fragments with the

protein solutions in 20 mM Tris pH 8 and 200 mM sodium chloride. The complexes were resolved on an 8% polyacrylamide gel running at 120 V and visualized using ChemiDoc MP Imaging System (BIORAD, CA, USA).

### Crystallization and crystal structure determination

Crystallization and data collection of the presented crystal structure have been reported previously[46]. Briefly, HigA2 and Opr17 were mixed in a 1:1.2 ratio to the final 250 μM concentration of the complex and subjected to crystallization trials using sitting drop vapor diffusion. Crystals grew over a several-day period in the 0.2 M sodium chloride, 0.1 M Na HEPES pH 7.5, 12% (w/v) PEG 8000 and were cryoprotected with an addition of 30% (w/v) glycerol. Data were collected at the SOLEIL synchrotron beamline Proxima-1 and XDS was used to index, integrate, and scale data[47]. Structures were solved via molecular replacement using PHASER (version v1.12-2829-00)[48]. The structure of the HigA2−Opr17 complex was solved using the HigA2 C-terminal domain (PDB 5J9I) as search model[22]. The resulting electron density clearly showed the presence of the DNA molecule, which was then built manually using Coot (version v0.98)[49]. Several rounds of model building and refinement were performed using phenix.refine (version v1.12-2829-00)[50]. The final refinement steps included TLS refinement, one chain per group.

### Small-angle X-ray scattering

All SAXS data were collected in the HPLC mode on the SWING beamline at the SOLEIL synchrotron (Gif-sur-Yvette, France). Samples (typically at 8 mg/ml in 20 mM TRIS pH 8 and 200 mM sodium chloride, for nucleoprotein complexes, 15 mM magnesium chloride was added) were injected into a Shodex KW 402.5-4 F column and run at 0.2 ml/min. Data was processed with Foxtrot[51] and programs from the ATSAS package (version 3.0.2)[52].

All simulations were performed in Xplor-NIH (version 2.49)[53], starting from the structure of the HigA2−Opr17 complex. Residues not resolved in the X-ray structure, including the disordered N-terminal tail, were added in Xplor-NIH, followed by minimization of the energy function consisting of standard geometric (bonds, angles, dihedrals, and impropers) and steric (van der Waals) terms.

For refinement against the experimental SAXS data, the positions of the DNA atoms and the structured protein regions were kept fixed, while residues 2−39 comprising the disordered N-terminus were given a full degree of freedom. The computational protocol comprised an initial simulated annealing step followed by side-chain energy minimization as described previously[54]. Briefly, in addition to the standard geometric and steric terms, the energy function includes a knowledge-based dihedral angle potential and the SAXS energy term incorporating the experimental data[55]. In order to simulate molecular ensembles of multiple conformers, several copies of the molecular system ($N = 1-5$) were refined simultaneously[54]. Truncated SAXS curves ($q < 0.3$ Å$^{-1}$) were used as the sole experimental input.

In each refinement run, 100 structures were calculated, and 10 lowest-energy solutions−representing the best agreement with the experimental data−were retained for the subsequent analysis. The agreement between experimental and calculated SAXS curves (obtained with the calcSAXS helper program, which is part of the Xplor-NIH package) was assessed by calculating the $\chi^2$:

$$\chi^2 = \frac{1}{n-1}\sum_{i=1}^{n}\left(\frac{I(q)_{calc,i}-I(q)_{exp,i}}{\delta I(q)_{exp,i}}\right)^2 \tag{1}$$

where $I(q)_{calc,i}$ and $I(q)_{exp,i}$ are the scattering intensities at a given $q$ for the calculated and experimental SAXS curves, $\delta I(q)_{exp,i}$ is an experimental error on the corresponding $I(q)_{exp,i}$ value, and $n$ is the number of data points defining the experimental SAXS curve.

## NMR spectroscopy

NMR spectra were acquired at 298 K on a VNMRS 800 MHz NMR spectrometer (equipped with a triple resonance HCN cold probe) or a Bruker Avance III HD 800 MHz spectrometer (fitted with a TCI cryoprobe). The data were processed in NMRPipe (version 9.8)[56] and analyzed in Cara (version 1.9.1)[57] or CCPNMR (version 2.5.2)[58]. The assignments of backbone resonances of the HigA2 homodimer were obtained from a standard set of 3D HNCACB, HN(CO)CACB, HNCO, HNCA, and HN(CO)CA experiments. The sample contained 250 μM U-[$^{13}$C,$^{15}$N] HigA2 (full protein sequence is given in Table S2) in 10 mM sodium acetate pH 4.0, 10% D$_2$O for the lock and 10 mM glutamate and 10 mM arginine to stabilize the protein. Titrations with DNA were performed in a different buffer due to precipitation upon the addition of DNA in low pH buffer. The NMR assignments were therefore transferred to the [$^1$H,$^{15}$N] HSQC spectrum of the protein sample in 20 mM sodium phosphate pH 6.8, 130 mM NaCl, and all experiments with DNA were conducted in this buffer. Threshold deviations from random-coil $^{13}$Cα, $^{13}$Cβ, $^{13}$CO, and $^1$Hα chemical shifts were calculated using the chemical shift index (CSI) module in CCPNMR[59], and the secondary structure of N-terminal part of HigA2 was predicted by the DANGLE module[60] in CCPNMR.

The NMR titration was performed by recording a set of 2D [$^1$H,$^{15}$N] HSQC spectra upon incremental addition of 1 mM Opr25 DNA stock solution to a U-[$^{15}$N] HigA2 sample at the initial concentration of 37.5 μM. The assignments of HigA2 backbone amide resonances in the Opr25-bound form were obtained from 3D BEST-HNCACB, HN(CO)CACB, HNCO, HN(CA)CO experiments on 320 μM U-[$^{13}$C,$^{15}$N] HigA2 sample with 2 molar equivalents of Opr25.

## Isothermal titration calorimetry

Samples were dialyzed against 20 mM sodium phosphate buffer pH 7.0, 150 mM sodium chloride, 1 mM EDTA, and filtered. Concentrations of proteins and DNA duplexes were determined by measuring the UV absorption and calculated using the estimated extinction coefficients for proteins and oligonucleotides[61]. Prior to the experiments samples were degassed for 20 min. Experiments were performed in the VP-ITC microcalorimeter (MicroCal, CT, with the VPViewer 1.4.12 software). Raw thermograms were exported and integrated with the NITPIC (version 1.1.0)[62] and then analyzed with SEDPHAT (version 15.2b)[63,64]. Parameter and error analysis were performed using the Monte-Carlo approach for nonlinear regression with 1000 iterations as implemented in SEDPHAT.

## Circular dichroism

Measurements were carried out using a J-1500 Circular Dichroism Spectrophotometer (JASCO, MD, USA, with the JASCO Spectra Manager software). Circular dichroism spectra were collected in 1 or 5 mm cuvette in the range 190-250 nm, using a spectral bandwidth of 0.5 nm and an averaging time of 2 s. Temperature scans were recorded between 4 and 94 °C by measuring the intensity at 222 nm in intervals of 1 °C at a speed of 1 °C/min. Measured signals (millimolar ellipticity) were then converted to mean residue molar ellipticity.

## In vivo repression

For the in vivo experiments, pSB1A3 vectors harboring wild-type HigA2 and its variants were transformed into competent *E. coli* BL21[DE3] cells. Overnight cultures were grown in 6 mL LBA at 37 °C with constant shaking at 120 rpm. Cultures were then diluted into fresh 6 mL LBA media at a ratio of 1:50 and further incubated at 30 °C with constant shaking at 120 rpm. OD600 and fluorescence intensity measurements were performed at 24 h. Six replicates of 100 μL for each sample were measured in a Greiner CELLSTAR® 96 well plate using plate reader Tecan Infinite 200 Pro with the Tecan i-control software ($\lambda_{ex}$ = 584 nm $\lambda_{em}$ = 630 nm). Promotor activity (PA) was determined by normalizing the fluorescence intensity corrected for the initial intensity (at 0 h) with the OD600.

## Statistics and reproducibility

No sample size calculation was performed, given that tested hypotheses do not refer to any particular population. No data was excluded from the presentation. For analysis of ITC thermograms, the first point is omitted in all cases, as per standard practice related to the technical operation of the instrument. For SAXS analysis, truncated data was used ($q < 0.3\,Å^{-1}$). DNA footprint shows (Fig. 1b) shows the result from a single independent experiment. Bar charts show the mean of at minimum three biological replicates ($n = 3$, bacterial cell cultures from different colonies) with error bars showing one standard deviation. Images of molecular models were created using UCSF Chimera (https://www.cgl.ucsf.edu/chimera/).

## Reporting summary

Further information on research design is available in the Nature Portfolio Reporting Summary linked to this article.

## Data availability

The crystallographic structural data for HigA2-Opr17 has been deposited into the Protein Data Bank under accession code 8A0W. The $^1$H, $^{13}$C, and $^{15}$N chemical shifts for HigA2 have been deposited in the Biological Magnetic Resonance Data Bank under Accession Number 51448. SAXS data has been deposited to SASDB under codes SASDS76 (HigA2) and SASDS86 (HigA2-Opr25). Source data are provided in this paper.

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

## Acknowledgements

This work was supported by grants from FWO-Vlaanderen (grant G003320N) and the Slovenian Research Agency (core funding P1-0201 and grant J1-50026). We acknowledge the support of the Center for Research Infrastructure at the University of Ljubljana, Faculty of Chemistry and Chemical Technology, supported by grant I0-0022 from the Slovenian Research Agency. We thank Prof. dr. Roman Jerala for the support with the initial in vivo experiments, dr. Marina Klemenčič for the assistance with the design of vectors and Domen Oblak for help with ITC experiments. The authors would also like to acknowledge help from the staff of the PROXIMA-1 and SWING beamlines (synchrotron SOLEIL, St-Aubin, France) for X-ray and SAXS data collection.

## Author contributions

S. Hadži, J.L., and R.L. designed the study. S. Hadži and R.L. wrote the paper with the help of A.V. and J.L. S. Hadži performed the crystallography studies, CD, ITC, and SAXS experiments together with the protein expression and purification. Z.Ž. performed in vivo experiments, EMSA, and prepared figures for the manuscript. M.K. and J.P. performed NMR experiments on free HigA2 and assigned NMR spectra. U.Z. prepared NMR samples and analyzed ITC data. D.C. performed copper phenanthroline footprinting. S. Haesaerts expressed and purified HigA2 protein variants. A.V. performed NMR titrations and analyzed NMR and SAXS data of the HigA2-operator complex.

## Competing interests

The authors declare no competing interests.
