## [Peer Review File · Nature Communications]

Fuzzy recognition by the prokaryotic transcription factor HigA2 from *Vibrio Cholerae*Reviewer #1 (Remarks to the Author):

In this manuscript, Hadzi and colleagues describe the interaction of antitoxin HigA2, and particularly its N-terminal intrinsically disordered region (IDR), with the toxin operator DNA. The authors determined the crystal structure of HigA2 bound to the operator DNA and used NMR, SAXS, ITC and mutagenesis approaches to characterize this interaction in detail. The manuscript reports novel findings, and the structural data will help to better understand the mechanism by which HigA2 acts, however disappointingly no biological or functional assays are included to validate this mechanism. Another weakness of the study is that only the structure of HigA2 bound to DNA is reported here. Adding the structure of the HigB2-HigA2-DNA complex that the authors have deposited to PDB (8A0X) but do not describe in the current manuscript would strengthen this work.

A few specific comments:

1. The IDR is unfortunately not resolved in the crystal structure, but this region and its binding to DNA is the major point of the study.
2. HigA2 forms a dimer in the crystal structure of the HigA2-DNA complex. Is this dimerization physiologically relevant or it's a crystallization artifact? Is dimerization required for binding to DNA? The dimerization interface needs to be characterized at least by mutagenesis.
3. The explanation of ITC results, described on one and a half pages (5-6) could be shortened into a few sentences.
4. Overall, it remains unclear as to why binding of HigA2 to DNA is fuzzy.
5. Unclear why NMR resonances of the globular domain of HigA2 disappear upon DNA binding. The resulting complex is not large and is within the detection limits of NMR.
6. Fig. 4c, the complete overlay of HSQC spectra of HigA2 (while DNA was titrated in) needs to be shown. The data shown in Fig. 4d raise more questions than answers.
7. A histogram of CSPs per HigA2 residue should be shown.
8. If IDR of HigA2 binds to DNA, this peptide needs to be assayed separately: its binding to DNA could be monitored by NMR and binding affinity could be measured by ITC or other quantitative methods.

Reviewer #2 (Remarks to the Author):

In this manuscript, the author explores the fuzzy binding of a prokaryotic intrinsically disordered protein (IDP) with RNA. The manuscript is well-written and the research is well-structured, employing various methods to effectively describe the fuzzy interactions. While I lack expertise in this specific area, the findings appear significant, although a more comprehensive assessment from an expert would be valuable.

Some of the techniques employed, such as DNA footprinting, EMSA, and in vivo repression, are unfamiliar to me. However, the other techniques used are appropriately applied and convincingly support the author's claims.

Overall, the manuscript is well-executed, and the research findings provide valuable insights into the fuzzy binding of the prokaryotic IDP with RNA.

Some minor points could be addressed to further improve the quality of the manuscript:

- In line 263-264, it is stated that "The 10 lowest-energy solutions fit well within the molecular

envelopes calculated from ab initio models with DAMMIF." However, it is not clear why this should be the case since ab initio models provide an average representation and may not perfectly match all conformations. It would be helpful to have this point clarified to better understand the relationship between the lowest-energy solutions and the DAMMIF envelopes.

- The NMR and MX data have been appropriately deposited in the relevant databases. It would be beneficial to also deposit the SAXS data in SASBDB for accessibility and data transparency.

Reviewer #3 (Remarks to the Author):

In this paper, Hadzi et al. present a thorough structural and biophysical analysis of the transcription factor activity of the *Vibrio cholerae* HigA2 antitoxin from the HigBA2 type II toxin-antitoxin system. Using in vivo, in vitro, and structural studies, the authors carefully unravel the relevance of the intrinsically disordered region of the HigA2 antitoxin for DNA recognition. This turns out to break with the current dogma in the IDP field, where the IDP sequence can be scrambled and still retain its function. The finding that this is not the case for HigA2 is important and opens for answering many new questions about the nature of the disordered regions of proteins in general and antitoxins from TA systems specifically.

The paper contains a number of significant, new insights and is overall of a high quality. It would therefore be suitable for publication following attention of the authors to the criticism raised below.

Major points

Figure 1. The EMSA assay needs to be improved. Since the control DNA is only run at 5, 10, and 15 μM and thus not in the range where binding takes place for the promoter DNA (1-2.5 μM), it is impossible to confirm the conclusion that the binding is weaker for the unrelated DNA. The experiment should be repeated with all samples on the same gel and using the full set of protein concentrations for both DNA fragments. It would also be easier to compare the results if the same length DNA probes were used. What are the extra (very strong) bands on the unrelated DNA lanes F and 5 μM ? Finally, it would be good to comment on the observation that the affinities measured by EMSA and ITC differ by several orders of magnitude.

Results, l. 160. Why do the altered enthalpy curve slopes necessarily imply hydrophobic interactions?

Results, l. 203-204. I believe that the effect could also be indirect, i.e. via modulation of the conformation of nearby (polar) residues when non-classical DNA-interacting residues (e.g. hydrophobic residues) are mutated.

Results, p. 8/l. 254. I am not sure you can conclude that the residues directly interact with DNA. It seems that some of these 'direct interactions' could be explained just as well by coordinating other residues for direct interaction?

Figure 2d and Figure 3. All data points (if less than 5-10) should be shown in all bar plots.

Minor points

Introduction, l. 48-49. Please introduce the Sic1 and Cdc4 proteins, and at l. 64-65, introduce the KID, KIX, and CREB domains.

Introduction, l. 63. "with its affinity is further enhanced", "is" missing?

Introduction, l. 71-76. What is the data supporting the IDPs are less frequent in bacteria given

that they are "heavy understudied"?

Introduction, p. 3/l. 79-80. Consider that not all antitoxins are IDPs, and using the definition in l. 84-85 might be more appropriate.

Introduction, p. 3/l. 84. "TA antitoxin" = "toxin-antitoxin antitoxin", consider simply "antitoxin".

Introduction, l. 87. "IDP action" is unclear, consider rephrasing.

Introduction, l. 89. "CcdB-poisoned", do you mean "CcdB-poisoned"?

Results, l. 103 and throughout. Figures are labelled "A, B, C..." in the text but "a, b, c..." in the figures.

Results, l. 124-131. Please list the interactions using the specific atoms involved, e.g. "hydrogen bond to N9 of the adenine base", etc.

Results, l. 134. How does HigA2 change upon DNA binding? Expand, contract, etc.

Results, l. 140 and throughout. The naming HigA2 Δ 37-104 implies that residues 37-104 have been removed whereas in fact it's 1-36. Consider renaming this variant.

Results, l. 147-148. "A titration of HigA2 into _a_ shorter DNA fragment"... "give identical affinity as with _the_ longer fragment...", i.e. articles missing.

Results, p. 7/l. 205. Please introduce the HigB2 toxin more carefully here as it is easy to mix up "HigA2" and "HigB2". For example, by writing "involved in HigB2 toxin contacts".

Results, l. 225. "While the latter", I think you mean "the first"? Also see below.

Results, l. 267. "The making group", what is meant by this?

Results, l. 272. The so-called "second group" of conformers should be clear from Figure 4E.

Results, l. 280. I don't understand how Thr+Lys can engage in hydrophobic interactions?

Discussion, l. 315. "the latter". I am not sure which of the two is referred to?

Table 1. Use same unit (nM?) for both temperature regimes for easier comparison?

Figure 1 legend. The definition of "+" in Figure 1b is missing.

Figure 2 legend, l. 401. "transfected" should be "transformed". Also, what does "ø" refer to?

Figure 1a. The schematic can be improved by indicating more clearly which part of the operon is zoomed in the inset.

Figure 2c. Please include R² values for each fit in the plot.

Figure 4a. When concluding that the IDP region only folds upon HigB2 binding (l.220), could the change in signal arise from the addition of HigB2?

Figure 5a. Wouldn't it be better to let 5a represent the unbound "native" situation, for which the entropic repulsion idea is still relevant, rather than scrambled DNA sequence? It's a bit difficult to understand when the top panel relates to a different situation than below.

Table S2. The space group listed, P3(1)2(1) does not match the one listed in the PDB validation summary report, P3(2)21 and there is (hopefully) a decimal point missing in the crystal-detector distance (509700 mm). Also, consider splitting macromolecule atoms into protein and DNA. Finally,

units are missing for some of the latter rows (RMS and B factors).

Table S3. "HigA2Shuff" is called "HigA2Ser2-37" in the text.

Figure S1c. Please consider doing a global alignment of the two structures rather than aligning on one side to better show the opening. Also, include the distance between the DNA-binding domains for both free and bound forms of HigA2.

Figure S1d. Please include Fo-Fc electron density for the dual Arg conformations as well as more precise indications of interactions, consistent with the text (p4-5/l.116-136). For example, which atom in G05 does R77 interact with?

Reviewer #1 (Remarks to the Author):

In this manuscript, Hadzi and colleagues describe the interaction of antitoxin HigA2, and particularly its N-terminal intrinsically disordered region (IDR), with the toxin operator DNA. The authors determined the crystal structure of HigA2 bound to the operator DNA and used NMR, SAXS, ITC and mutagenesis approaches to characterize this interaction in detail. The manuscript reports novel findings, and the structural data will help to better understand the mechanism by which HigA2 acts, however disappointingly no biological or functional assays are included to validate this mechanism.

While our work indeed focusses on biophysics and understanding the molecular mechanism behind fuzzy protein-DNA recognition, we did in fact include a number of functional/biological assays that support our findings related to the novel function of HigA2 IDR. In addition to thorough biophysical and structural characterization of IDR (done *in vitro*) we also performed *in vivo* reporter assay with numerous variants of HigA2 and *in vivo* full alanine scan of HigA2 IDR. We specifically designed the reporter system on the low copy number plasmids and we used weak promoters in order to mimic the real situation as closely as possible.

Furthermore, based on the reviewer's comments we now also include an additional functional assay, which addresses the functional trade-off of the IDR, that is the neutralization of HigB2 vs. interactions with the operator. Specifically, we performed bacterial spotting for a selection of alanine mutants in order to understand whether any of these mutants also impair toxin activity. We observed that the selected single-residue mutants do not reduce neutralization capacity of HigA2 and therefore indicate that these conservation in the specific IDR region might be related to its operator interactions, rather than toxin neutralization. This bacterial spotting assay is now described in the manuscript and the data is shown on Figure S5.

Another weakness of the study is that only the structure of HigA2 bound to DNA is reported here. Adding the structure of the HigB2-HigA2-DNA complex that the authors have deposited to PDB (8A0X) but do not describe in the current manuscript would strengthen this work.

We understand that the availability in the protein data bank of this additional structure immediately raises the question as why it was omitted from the manuscript. In fact, in our initial drafts, we tried to include it in our story. We noticed, however, that this resulted in a manuscript without clear focus and that it also significantly lengthened it. We decided that it would be best to present a manuscript with a clear focus (fuzzy protein-DNA recognition) while omitting side stories that make the paper more confusing and difficult to follow. We think in the end that the HigB2-HigA2-DNA structure is not so relevant as it is likely to be considered as an artefact. Indeed, while we can get the complex associate and crystallize at concentrations that are not relevant *in vivo*, it will never form in its natural context.

A few specific comments:

1. The IDR is unfortunately not resolved in the crystal structure, but this region and its binding to DNA is the major point of the study.

This is correct, and in fact is in this context an important observation rather than just a negative one. That it is not visible while still contributing to binding affinity is what points towards fuzzy interactions and is also the reason why such interactions are difficult to characterize. We therefore applied a pallet of different, complementary structural (CD, SAXS, NMR, alanine scanning) methods to characterize this fuzzy interaction, which gave us, we believe, a very coherent model of this IDR-DNA interaction.

2. HigA2 forms a dimer in the crystal structure of the HigA2-DNA complex. Is this dimerization physiologically relevant or it's a crystallization artifact? Is dimerization required for binding to DNA? The dimerization interface needs to be characterized at least by mutagenesis.

As shown in our previous publications, like the overall majority of TA antitoxins, HigA2 exists as a dimer in solution as well as in the crystal. In general, independent of any specific TA family, such dimerization is well known to be essential for DNA binding but not for toxin neutralization. This has perhaps not been emphasized sufficiently in our manuscript, therefore we now explicitly mention the dimeric nature of HigA2 referring to our previous publications.

As the reviewer points out, the structure of HigA2-DNA indicates that dimeric HigA2 is required for binding the operator and suggests a control experiment to verify the hypothesis whether dimerization is required for DNA binding. As suggested we performed mutagenesis and introduced two mutations L87D and T97D, which introduce a negative charge at the dimer interface and disrupt HigA2 dimerization. We show that while wt HigA2 can repress mRFP1 transcription from the native higBA2 operon, while the mutant with impaired dimerization cannot. Given that HigA2 is obligate dimer in solution (and crystal) and that L87D and T97D mutants introduced at dimer interface impair DNA-binding, we conclude that dimerization is physiologically relevant and is required for DNA binding.

Based on the reviewer's comment we now refer to previous data showing that HigA2 is an obligate dimer in solution and include the new mutagenesis results in the manuscript (**Figure 2d, shown below**).

3. The explanation of ITC results, described on one and a half pages (5-6) could be shortened into a few sentences.

The difference in affinities determined by ITC between the HigA2 and its truncated version without IDR gave us the very first indication regarding the importance of the IDR for DNA binding. Further ITC experiments with different HigA2 mutants gave valuable information on the nature of this interaction such as positional dependence of interactions, resistance to sequence shuffling and importance of hydrophobic interactions. All these ITC results agree well with the *in vivo* and structural data and give a complementary picture of IDR-DNA interaction.

We tried to shorten this section however, due to the many different mutations used, the shortened version was difficult for reader to follow. Indeed, the ITC results are described at length, however we feel that they are important and that drastically shortened description of all ITC data will be impossible to follow. We nevertheless tried our best and shortened the section describing ITC results to some extent from 3 to 2 paragraphs in the section titled "*The HigA2 intrinsically disordered domain enhances operator binding*."

4. Overall, it remains unclear as to why binding of HigA2 to DNA is fuzzy.

Why it needs to be fuzzy is a philosophical question: the same outcome of transcription regulation can be achieved via a variety of mechanisms, and in this particular case it evolved via fuzzy interactions. It might be very interesting to study how it evolved, but that brings us far from our core question as how does it work. The latter is also difficult to answer since there is not a single experimental method available which would give precise, atomic-resolution picture of fuzzy interactions. For this reason, different complementary methods, each giving one detail need to be combined to gain the full picture, although sometimes even this will not give the same level of information as a classical crystal structure of a protein-protein complex.

HigA2 can also form non-fuzzy interaction, for example when IDR binds the toxin HigB2 it folds-upon-binding and gives a well-ordered protein complex, that can be crystalized and studied in detail. It is therefore interesting that the same IDR can also engage in a different, more dynamic interaction, which we believe it ultimately boils down to the basic issue of the balance between entropy vs. enthalpy. As we show the default behavior of IDR is to resist any interactions due to entropically unfavorable confinement. When IDR can form weak interactions (small gain in enthalpy to compensate entropy) such IDR will remain predominantly dynamic, as in the HigA2-DNA complex. On the other hand, when IDR can form strong interactions with the target, this will be sufficient to overcome entropic penalty and a well-ordered, non-fuzzy complex is formed, as in HigA2-HigB2 complex.

5. Unclear why NMR resonances of the globular domain of HigA2 disappear upon DNA binding. The resulting complex is not large and is within the detection limits of NMR.

While the increase in the molecular weight of the system upon DNA binding is significant (22 kDa for the free HigA2 versus 39 kDa for the HigA2-DNA complex), taken at the face value the overall size is still within the NMR detection limit as pointed out by the reviewer. However, compared to a globular macromolecule of the same size, the HigA2-DNA complex (featuring a globular HigA2 domain interacting with a nearly-linear DNA double-helix, Fig. 1c and Fig. 4e) will have larger hydrodynamic radius and, thus, larger rotational correlation time (τ_c). Also the IDR may contribute to a slower tumbling even if IDR resonances remain visible due to local dynamics. In other words, the HigA2-DNA

complex will tumble much slower in solution than a globular protein of the same size, which would lead to a much faster T2 relaxation, explaining the rapid signal loss. Contrary to the ordered part of the HigA2-DNA complex (tumbling as a single unit), a flexible IDP tail exhibits a much higher degree of freedom, resulting in smaller τ_c and, hence, slower relaxation, allowing to detect the NMR signal.

Incidentally, we see the same behaviour in other protein-DNA complexes studied in our lab, e.g. YdaS-DNA (unpublished results) or Rec114-Mei4-DNA (Daccache et al. *Genes Dev.* 2023, 37, 535-53): NMR signals of the globular protein domain disappear upon DNA binding and different mechanisms may be at play (intermediate exchange, tumbling rate, ..).

6. Fig. 4c, the complete overlay of HSQC spectra of HigA2 (while DNA was titrated in) needs to be shown. The data shown in Fig. 4d raise more questions than answers.

A complete overlay of HSQC spectra for HigA2 with different DNA molar ratios is now shown in Figure S7. As the figure shows, cross-peak intensity starts to decrease upon addition of DNA and only a handful of signals remain at 2 molar equivalents of DNA. Most of these remaining peaks were explicitly assigned using 3D NMR experiments. Given that these signals were assigned in both free and bound states the observed chemical shifts difference is reliable and strongly supports a direct interaction between IDR and operator.

We would nevertheless prefer to keep Figure 4c as it is (only 2 HSQC spectra are overlaid- unbound and fully bound HigA2), since in the complete overlay of HSQCs the changes are difficult to see. For this reason, we also keep Figure 4d, as it more clearly shows the changes for selected residues change during the full course of DNA titration. Note that in Figure 4d there is no incremental shift during the titration, because HigA2-DNA binding is in slow exchange on the NMR chemical shift timescale. Rather, the peaks of the free HigA2 are progressively decreasing in intensity, then peaks of the bound HigA2 appear somewhere else in the spectrum and keep on increasing in intensity as the titration progresses. This is a classical NMR book example of the slow exchange, which is what Fig. 4d shows. As there is no progressive “shift” in the titration, we could not simply follow the peaks moving to a new position, but had to explicitly assign the DNA-bound form (red in Fig. 4c) using a set of 3D experiments, just as was done for the free HigA2.

Based on the reviewer's comment we now show a complete overlay of HSQC spectra in Figure S7, as shown below.

Shown on above is the new Figure S7. **Titration of operator DNA to HigA2.** a) Overlay of eight ^1H - ^{15}N HSQC spectra of HigA2 protein with various molar equivalents of 25 bp promoter DNA (Opr25). Titration was conducted on 37.5 μM HigA2 protein at pH 6.8 and 25 $^\circ\text{C}$. Sample contained 20 mM sodium phosphate buffer and 130 mM NaCl. Spectra were recorded on 800 MHz NMR spectrometer. b) Histogram of the average chemical shift perturbations for backbone atoms between free HigA2 and HigA2 with 2.0 molar equivalents of Opr25 DNA. The resonances of these residues were explicitly assigned in the both free and bound HigA2.

7. A histogram of CSPs per HigA2 residue should be shown.

A histogram of CSP per HigA2 residue is now included in SI7 together with the complete overlay of HSQC spectra. Only 10 residues were visible and could be assigned in HigA2-bound form. The average chemical shift perturbations ($\Delta\delta_{avg}$) were calculated as $\Delta\delta_{avg} = (\Delta\delta N/50 + \Delta\delta H/2)0.5$, where $\Delta\delta N$ and $\Delta\delta H$ are the chemical shift changes of the backbone amide nitrogen and proton, respectively.

Based on the reviewer's comment we calculate CSP per HigA2 residue and include the histogram in Figure S7, as shown above in the reply to question 6.

8. If IDR of HigA2 binds to DNA, this peptide needs to be assayed separately: its binding to DNA could be monitored by NMR and binding affinity could be measured by ITC or other quantitative methods.

The presence or absence of the IDR changes the HigA2-DNA binding constant by a factor less than 100. If we assume that the interactions established by the IDR and the folded domain are additive, the binding constant for the IDR on its own should be $\gg 10$ mM and therefore not readily detectable *in vitro* (a.o. due to limitations of peptide solubility). But still, the suggested experiment would serve as a good control. We used two independent methods to assess IDR binding to DNA operator: ITC and *in vivo* repression. In the ITC experiment we titrated HigA2 peptide (residues 3-39 corresponding to the IDR, named hereafter as HigA2_{IDR}) to the DNA operator (Opr25). The observed heats of injection are very small and do not change with the molar ratio, which is indicative of dilution effects and lack of binding (shown below on left). In the second experiment we performed *in vivo* repression assay, where HigA2_{IDR} was expressed under a constitutive promoter and repression was monitored via fluorescence of mRFP1 under native *higBA2* promoter. Fluorescence of cells harboring this vector (last bar on right HigA2_{IDR}) is similar to the cells harboring only *higBA2* mRFP1 (positive control), indicating that IDR region alone is unable to repress i.e. bind to the *higBA2* operon.

We conclude that the HigA2 IDR alone is insufficient to bind the operator, rather it needs to be attached to the globular domain. This is in accordance with our model, where the globular HigA2 domain serves to locate the correct operator sequence and provide the majority of binding free energy, while the IDR enhances binding. Without the globular domain, which places IDR close to the operator (increase in the local IDR concentration), IDR has too weak affinity for the operator to be able to bind by itself.

These results are now reported in the manuscript and shown in Figure 2d (*in vivo* repression), Figure S4 (ITC thermogram) and Table 1 (ITC parameters).

Reviewer #2 (Remarks to the Author):

In this manuscript, the author explores the fuzzy binding of a prokaryotic intrinsically disordered protein (IDP) with RNA. The manuscript is well-written and the research is well-structured, employing various methods to effectively describe the fuzzy interactions. While I lack expertise in this specific area, the findings appear significant, although a more comprehensive assessment from an expert would be valuable.

Some of the techniques employed, such as DNA footprinting, EMSA, and in vivo repression, are unfamiliar to me. However, the other techniques used are appropriately applied and convincingly support the author's claims.

Overall, the manuscript is well-executed, and the research findings provide valuable insights into the fuzzy binding of the prokaryotic IDP with RNA.

We thank the reviewer for this generally positive response. Please find the answers to specific issues below.

Some minor points could be addressed to further improve the quality of the manuscript:

- In line 263-264, it is stated that "The 10 lowest-energy solutions fit well within the molecular envelopes calculated from ab initio models with DAMMIF." However, it is not clear why this should be the case since ab initio models provide an average representation and may not perfectly match all conformations. It would be helpful to have this point clarified to better understand the relationship between the lowest-energy solutions and the DAMMIF envelopes.

This is indeed correct: for an IDP we do not expect a close match to the ensemble and a DAMMIF envelope and therefore we think this is a significant feature to be mentioned. In our ensemble of DNA-bound HlgA2, the IDR does not adopt random conformations, but is restricted in its conformations and orientations to remain close to and parallel to the DNA. Thus, the IDR fits within a specific defined volume and one can therefore expect that this is reflected at least to some extent in a DAMMIF envelope (which serves as a further confirmation of our model). Upon a closer inspection it is evident that in few models the tip of IDR reaches out of the DAMMIF envelopes, as is also expected given that the conformational ensemble is restricted but not a folded structure. The majority of models are within the envelope, which is, it appears that the DAMMIF envelope in this case also captures the overall structure of the ensemble.

We slightly re-phrased this sentence which now reads: "*The 10 lowest-energy solutions generally fit within the molecular envelopes calculated from ab initio models with DAMMIF (Figure S8).*" "

- The NMR and MX data have been appropriately deposited in the relevant databases. It would be beneficial to also deposit the SAXS data in SASBDB for accessibility and data transparency.

This was done, all SAXS data were submitted to SASDB (<https://www.sasbdb.org>) under codes SASDS76 for Antitoxin HlgA-2 and SASDS86 for the HlgA-2-DNA complex. This is now mentioned under the section 'Data availability'.

Reviewer #3 (Remarks to the Author):

In this paper, Hadzi et al. present a thorough structural and biophysical analysis of the transcription factor activity of the *Vibrio cholerae* HigA2 antitoxin from the HigBA2 type II toxin-antitoxin system. Using in vivo, in vitro, and structural studies, the authors carefully unravel the relevance of the intrinsically disordered region of the HigA2 antitoxin for DNA recognition. This turns out to break with the current dogma in the IDP field, where the IDP sequence can be scrambled and still retain its function. The finding that this is not the case for HigA2 is important and opens for answering many new questions about the nature of the disordered regions of proteins in general and antitoxins from TA systems specifically.

The paper contains a number of significant, new insights and is overall of a high quality. It would therefore be suitable for publication following attention of the authors to the criticism raised below.

We thank the reviewer 3 for recognizing the uniqueness in sequence specificity observed in this fuzzy interaction. We are particularly grateful for very detailed revision of the manuscript and drawing our attention to number of issues which were corrected and improved to best of our ability as outlined in the response below.

Major points

Figure 1. The EMSA assay needs to be improved. Since the control DNA is only run at 5, 10, and 15 μ M and thus not in the range where binding takes place for the promoter DNA (1-2.5 μ M), it is impossible to confirm the conclusion that the binding is weaker for the unrelated DNA. The experiment should be repeated with all samples on the same gel and using the full set of protein concentrations for both DNA fragments. It would also be easier to compare the results if the same length DNA probes were used. What are the extra (very strong) bands on the unrelated DNA lanes F and 5 μ M? Finally, it would be good to comment on the observation that the affinities measured by EMSA and ITC differ by several orders of magnitude.

We agree. The experiment was repeated exactly as suggested by the reviewer, that is we used identical concentration range 0-10 μ M of HigA2, all three DNA fragments are of exactly the same length (operator region, intergenic region and unrelated DNA). Also specific (operator region) and nonspecific binding (intergenic or random DNA) are now shown on the same gels.

In the repeated experiment there are no "extra" DNA bands as were present previously. These probably corresponded to a small amount of single-stranded DNA that contaminated our sample. Also, in the repeated experiment, we observe no significant differences in the HigA2-operator binding affinities between EMSA and ITC. Binding to the DNA already occurs at lowest concentration (0.1 μ M), close to what would be expected based on the affinity determined from ITC. In the previous EMSA this was not evident, likely because of lower detection limit (previous EMSA in SI1a was stained using EtBr, while now we used fluorescently labeled DNA fragments which achieve lower detection threshold).

Overall, the improved experiment now supports our previous conclusions more clearly, showing that HigA2 binds specifically to the operator region, while binding to the intergenic region is due to nonspecific DNA binding at higher concentrations. We therefore conclude that there is a single operator regulating *higBA2*, one that is upstream of *higB2* gene. In SI we now replace the old figure with the new one shown below.

Results, l. 160. Why do the altered enthalpy curve slopes necessarily imply hydrophobic interactions?

Heat capacity changes in macromolecular binding (also folding) arise from the change in solvation of hydrophobic residues (ref ref). When hydrophobic residues are partially de-solvated or buried inside the complex, this leads to negative change in heat capacity as a consequence of hydrophobic effect. We observe that HigA2 and HigA2_{ΔIDR} have different slopes on ΔH vs T plot, therefore their heat capacity change for Opr45 binding is different ($d\Delta H/dT = \Delta C_p$). Given that the magnitude of heat capacity change scales linearly with the burial of hydrophobic surface area, we therefore concluded that less hydrophobic surface is buried upon formation of HigA2_{ΔIDR}-operator complex.

To clarify this in the main text we rephrased the sentence in lines 160 which now reads:

"The binding enthalpy decreases linearly with temperature, indicating a negative heat capacity change (Figure 2c), which scales with the amount of hydrophobic surface buried upon binding (Spolar & Record PNAS 1989). Given that HigA2_{ΔIDR} has lower heat capacity (lower slope) this suggests that removal of IDR also removes hydrophobic contacts mediated by the IDR."

Reference: Spolar, R S et al. "Hydrophobic effect in protein folding and other noncovalent processes involving proteins." Proceedings of the National Academy of Sciences of the United States of America vol. 86,21 (1989): 8382-5.

Results, l. 203-204. I believe that the effect could also be indirect, i.e. via modulation of the conformation of nearby (polar) residues when non-classical DNA-interacting residues (e.g. hydrophobic residues) are mutated.

Yes this could also be possible. Strictly speaking there is now way to tell from any experiment whether the effect arises from direct or indirect effect. The effects from mutations do seem to appear in clusters (the N-terminal part and the segment between residues 24-32, which suggests a concerted event. Also, some complementary experiments seem to point towards involvement of hydrophobic residues in the IDR-DNA interaction, for example some models in the SAXS-derived ensemble and also the difference in heat capacities between HigA2 and HigA2_{ΔIDR}} binding to DNA. We nevertheless decide to include an additional explanation concerning indirect effects (underlined sentence in the paragraph below):

"Thus, both electrostatic and hydrophobic residues seem to mediate IDR-DNA contacts. Of these residues Lys24, Thr26, Leu27 and partially also Lys28 are conserved in HigA2 homologs, while Leu25 is variable (Figure S5). Leu27, Lys28 and Val32 are involved contacts with HigB2 toxin but they are not essential for toxin neutralization, suggesting that the evolutionary conservation may be related to fuzzy DNA binding. Next to direct interaction of the hydrophobics with the DNA, they may also affect the structural ensemble and modulate the conformation of nearby (polar) residues."

Results, p. 8/1. 254. I am not sure you can conclude that the residues directly interact with DNA. It seems that some of these 'direct interactions' could be explained just as well by coordinating other residues for direct interaction?

Yes, this is probably more precise, which is also how we interpret chemical shift perturbations in the sentences proceeding the one that refers to "direct interactions" ("*It is likely that the residues with detected NMR binding shifts do not interact directly with the DNA. Rather they experience secondary binding effects propagating from the neighboring groups which pull them towards the DNA, thereby altering their chemical environment. Such indirect binding effects could explain why, unlike for residues involved in direct interactions with the DNA, the affected resonances can be observed in the DNA-bound form and account for the modest chemical shift changes seen in the HSQC spectrum (Figure 4c).*").

In line with what we wrote and reviewer's comments we think it would be better to rephrase the last sentence and attribute these "direct" interactions to the IDR as a whole (given that the mentioned residues are more likely to experience secondary shifts). As we explain in the manuscript the IDR residues with do make 'direct' interactions most likely also loose the signal intensity and are therefore not observed with NMR (including globular domain residues). The last sentence now reads:

"Overall, the observed NMR chemical shift perturbations confirm that the HigA2 N-terminal domain engages in direct interactions with the operator DNA".

Figure 2d and Figure 3. All data points (if less than 5-10) should be shown in all bar plots.

This is now corrected, and all data points are included in bar plots in Figs 2 and 3.

Minor points

Introduction, l. 48-49. Please introduce the Sic1 and Cdc4 proteins, and at l. 64-65, introduce the KID, KIX, and CREB domains.

Sic1/Cdc4 and KID, KIX and CREB domains are now briefly introduced.

Introduction, l. 63. "with its affinity is further enhanced", "is" missing?

Indeed, this was corrected.

Introduction, l. 71-76. What is the data supporting the IDPs are less frequent in bacteria given that they are "heavy understudied"?

Basically, this comes from a literature search where one finds very few hits on studies involving IDPs in prokaryotes. Except for the work on bacterial antitoxins, we are aware of only two bio-informatics papers that specifically look for the occurrences of IDRs in prokaryote proteomes (Pavlović-Lažetić et al. BMC Bioinformatics (2011), 12:66 and Mitić et al. BMC Bioinformatics (2018) 19:158) next to a very small number of papers dealing with a few individual proteins. So, it may be correct to state that they are understudied relative to eukaryote IDRs. Prokaryotes also lack typical classes of functional IDRs that are abundant in eukaryotes such as transcription activation domains. The lower abundance of IDRs in prokaryotes (and their comparatively shorter lengths) has been noted before, see for example:

- Dunker AK et al. Intrinsic protein disorder in complete genomes. Workshop Genome Inform 2000;11:161–71.
- Ward JJ et al. Prediction and functional analysis of native disorder in proteins from the three kingdoms of life. J Mol Biol 2004;337:635–45, <http://dx.doi.org/10.1016/j.jmb.2004.02.002>.
- Oates ME et al. D2 P2 : database of disordered protein predictions. Nucleic Acids Res 2013;41:D508–16, <http://dx.doi.org/10.1093/nar/gks1226>.
- Peng Z et al. Exceptionally abundant exceptions: comprehensive characterization of intrinsic disorder in all domains of life. Cell Mol Life Sci CMLS 2014, <http://dx.doi.org/10.1007/s00018-014-1661-9>.

Based on the reviewer's comment we added the two most relevant references from above to support our claim in the text.

Introduction, p. 3/l. 79-80. Consider that not all antitoxins are IDPs, and using the definition in l. 84-85 might be more appropriate.

We changed the phrasing to:

"Among the most extensively studied prokaryotic IDPs are several representatives of antitoxins from the so-called toxin-antitoxin (TA) modules."

Introduction, p. 3/l. 84. "TA antitoxin" = "toxin-antitoxin antitoxin", consider simply "antitoxin".

This was corrected.

Introduction, l. 87. "IDP action" is unclear, consider rephrasing.

This was changed into:

"In some instances, function of IDP antitoxins extends from their main activity as a toxin inhibitors."

Introduction, l. 89. "CcdB-poisoned", do you mean "CcdB-poisoned"?

Yes, we corrected this typo.

Results, l. 103 and throughout. Figures are labelled "A, B, C..." in the text but "a, b, c..." in the figures.

This is true, we corrected all the references to the figures throughout the text.

Results, l. 124-131. Please list the interactions using the specific atoms involved, e.g. "hydrogen bond to N9 of the adenine base", etc.

This was corrected and interactions to specific nucleobase atoms are now described in the text.

Results, l. 134. How does HigA2 change upon DNA binding? Expand, contract, etc.

We corrected this sentence which now reads: *"In addition to structural changes of the DNA molecule, HigA2 slightly contracts upon DNA binding"*.

Results, l. 140 and throughout. The naming HigA2 Δ 37-104 implies that residues 37-104 have been removed whereas in fact it's 1-36. Consider renaming this variant.

Indeed, the name of this variant is rather confusing. Based on the editor's suggestion we now refer to IDP as IDR, therefore the HigA2 variant without IDR is now termed HigA2 Δ IDR. This has been now changed throughout the manuscript.

Results, l. 147-148. "A titration of HigA2 into _a_ shorter DNA fragment"..."give identical affinity as with _the_ longer fragment...", i.e. articles missing.

This was corrected.

Results, p. 7/l. 205. Please introduce the HigB2 toxin more carefully here as it is easy to mix up "HigA2" and "HigB2". For example, by writing "involved in HigB2 toxin contacts".

This was corrected as suggested.

Results, l. 225. "While the latter", I think you mean "the first"? Also see below.

This was corrected for clarity. The sentence now reads: *" While free HigA2 shows typical features of a spherical particle with a disordered tail, the plot of the HigA2-operator complex is closer to what is observed for globular particles, suggesting reduced flexibility of the disordered tail (Figure 4b)."*

Results, l. 267. "The making group", what is meant by this?

This was a typo and was corrected. The sentence now reads: "*For the first, larger group of conformers, the segment between the residues 25 to 37 runs parallel to the DNA helix at a distance of ~5-10 Å from the DNA backbone, while the N-terminal part of the chain extends out of the globular core (Figure 4e, representative model shown in violet).*"

Results, l. 272. The so-called "second group" of conformers should be clear from Figure 4E.

We now improved Figure 4e by showing a representative structure of two conformers in different colors.

Results, l. 280. I don't understand how Thr+Lys can engage in hydrophobic interactions?

This sentence has been slightly rephrased to improve meaning. What was meant was that the segment *between the residues Lys24 to Thr29* fits into the minor groove forming polar and hydrophobic interactions with the DNA. The sequence for this segment is KLTLKT, where the two leucine residues make hydrophobic contacts as depicted on Figure 4g. We now change the main text to:

"In one of the models the segment between residues 24 to 29 fits into the DNA major groove mediating both charged (Lys24 to phosphate backbone) and hydrophobic contacts (Leu25 and Leu27 to thymine methyl group) (Figure 4g).

Discussion, l. 315. "the latter". I am not sure which of the two is referred to?

We corrected this and the sentence now reads:

"The default behavior of an IDP segment attached to a globular domain is to prevent its association with potential interaction partners, or at least strongly weaken this interaction (Figure 5a)"

Table 1. Use same unit (nM?) for both temperature regimes for easier comparison?

This was corrected, all affinities are now reported in units of nM.

Figure 1 legend. The definition of "++" in Figure 1b is missing.

The symbols + and ++ were used to label HigA2-bound DNA, where HigA2 is at 1 (+) or 5 μM (++) concentration. These concentrations are now indicated on the Figure and replace the + and ++ symbols.

Figure 2 legend, l. 401. "transfected" should be "transformed". Also, what does "ø" refer to?

This was corrected and the "ø" symbols was replaced for "no mRFP1", which is a negative control (cells harboring an empty vector i.e. the one not expressing mRFP1).

Figure 1a. The schematic can be improved by indicating more clearly which part of the operon is zoomed in the inset.

Schematic was improved and the dashed lines now indicate more clearly the part of the operon which is zoomed.

Figure 2c. Please include R² values for each fit in the plot.

This was corrected and R² values are now shown next to the linear fits in Figure 2c.

Figure 4a. When concluding that the IDP region only folds upon HigB2 binding (l.220), could the change in signal arise from the addition of HigB2?

In principle yes, however the crystal structures of HigB2 without and with the HigA2 (PDBs 5MJE and 5JAA) show practically no change in the secondary structure content of HigB2 in both forms. Therefore, the change in the CD signal can be confidently attributed to the IDR.

Figure 5a. Wouldn't it be better to let 5a represent the unbound "native" situation, for which the entropic repulsion idea is still relevant, rather than scrambled DNA sequence? It's a bit difficult to understand when the top panel relates to a different situation than below.

We changed this figure to represent the wild-type protein, as this indeed makes more sense.

Table S2. The space group listed, P3(1)2(1) does not match the one listed in the PDB validation summary report, P3(2)21 and there is (hopefully) a decimal point missing in the crystal-detector distance (509700 mm). Also, consider splitting macromolecule atoms into protein and DNA. Finally, units are missing for some of the latter rows (RMS and B factors).

Yes, indeed these were wrong, and we corrected accordingly. Also missing units were added and macromolecule atoms are now listed separately for DNA and protein.

Table S3. "HigA2Shuff" is called "HigA2Ser2-37" in the text.

This was corrected and we now use HigA2_{Shuff} throughout the text.

Figure S1c. Please consider doing a global alignment of the two structures rather than aligning on one side to better show the opening. Also, include the distance between the DNA-binding domains for both free and bound forms of HigA2.

This was changed, the new Figure S2 now shows the global alignment of apo HigA2 and DNA-bound HigA2 with distances between the two DNA-binding domains indicated for each structure.

Figure S1d. Please include Fo-Fc electron density for the dual Arg conformations as well as more precise indications of interactions, consistent with the text (p4-5/l.116-136). For example, which atom in G05 does R77 interact with?

This was corrected, new Figure S2 now shows Fo-Fc electron density for both Arg77 conformations.

Reviewer #1 (Remarks to the Author):

The authors have adequately addressed my previous comments.

Reviewer #2 (Remarks to the Author):

The author appropriately addressed the minor points raised during the first review.

Reviewer #3 (Remarks to the Author):

The authors have carefully considered and implemented all of my suggestions and I fully support publication of the revised manuscript.